# Twin: Tuning Learning Rate and Weight Decay of Deep Homogeneous Classifiers without Validation

**Lorenzo Brigato**,* **Stavroula Mougiakakou**
*ARTORG Center, University of Bern*

**Reviewed on OpenReview:** *https://openreview.net/forum?id=1SIP2M2HJa*

## Abstract

We introduce **T**une **w**ithout Validat**ion** (Twin), a simple and effective pipeline for tuning learning rate and weight decay of homogeneous classifiers without validation sets, eliminating the need to hold out data and avoiding the two-step process. Twin leverages the margin-maximization dynamics of homogeneous networks and an empirical scaling law that links training and test losses across hyper-parameter configurations. This mathematical modeling yields a regime-dependent, validation-free selection rule: in the *non-separable* regime, training loss is monotonic in test loss and therefore predictive of generalization, whereas in the *separable* regime, the parameters' norm becomes a reliable indicator of generalization due to margin maximization. Across 37 dataset-architecture configurations for image classification, we demonstrate that Twin achieves a mean absolute error of 1.28% compared to an *Oracle* baseline that selects HPs using test accuracy. We demonstrate Twin's benefits in scenarios where validation data is scarce, such as small-data regimes, or difficult and costly to collect, as in medical imaging. Code available at https://github.com/lorenzobrigato/twin.

## 1 Introduction

Like most machine learning models, deep networks are configured by a set of hyper-parameters (HPs) whose values must be carefully chosen and which often considerably impact the final outcome (Krizhevsky et al., 2012; He et al., 2019; Yu & Zhu, 2020; Franceschi et al., 2024). Setting up wrong configurations translates into bad performance, particularly in the most difficult optimization scenarios, e.g., large models that overfit on small datasets (Lorraine et al., 2020; Brigato et al., 2021; 2022).

Despite their importance, HP tuning pipelines remain surprisingly cumbersome or data-inefficient. Traditionally, HP search is performed in two ways, as exemplified in Figure 1 (left and center). When a validation set is unavailable, practitioners must split the training set, which requires a cumbersome two-step process involving tuning and retraining. If the dataset is small, the resulting validation subset may yield noisy and unreliable HP estimates (Lorraine et al., 2020; Brigato & Mougiakakou, 2023). Alternatively, if a validation set is available, data must be collected or withheld upfront (typically 10–30%), which is costly or prohibitive in data-scarce domains such as medical imaging (Varoquaux & Cheplygina, 2022) or federated learning (McMahan et al., 2017). Although other methodologies, such as multi-

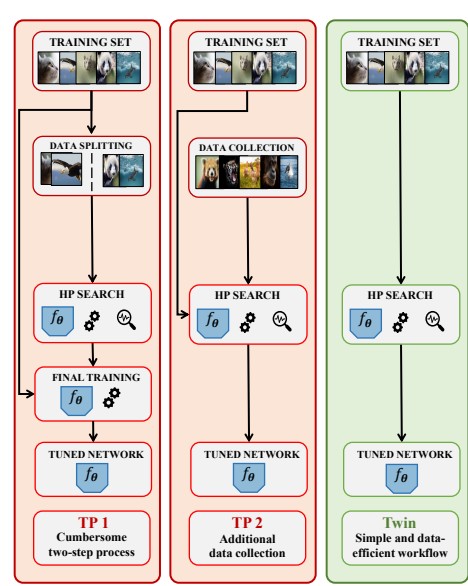

Figure 1: **Overview.** Twin simplifies the tuning workflow by avoiding the cumbersome two-step process or additional data collection of traditional pipelines (TPs).

---

*Corresponding author email: lorenzo.brigato@unibe.ch

fold or multi-round cross-validation (Stone, 1974), exist, they are scarcely employed in deep learning due to their computational overhead. We argue that splitting or additionally reserving data is avoidable.

Existing lines of work only partially address our problem. Validation-free early-stopping methods (Forouzesh & Thiran, 2021) monitor training trajectories to determine when to halt optimization, but they operate *within* a single run and do not offer principled cross-trial comparisons. Generalization-gap predictors require supervised meta-training or calibration across heterogeneous HP draws, limiting their reliability as ranking metrics (Jiang et al., 2018). Sharpness-based criteria fail to reliably correlate with generalization beyond specific architectures (Andriushchenko et al., 2023). Finally, current validation-free HPO (Mlodozeniec et al., 2023) have primarily focused on learning invariances but have struggled to scale beyond simple tasks (Schwöbel et al., 2022) and modest network sizes (Immer et al., 2022), thereby limiting their adoption.

Motivated by these challenges, we introduce **T**une **w**ithout Validat**ion** (Twin), a simple and innovative HP selection approach enabling practitioners to directly select the learning rate (LR) and weight decay (WD) of classifiers from the training set, as sketched in Figure 1 (right). Our approach builds upon two complementary insights: **(1)** the theoretically grounded **margin-maximization of homogeneous classifiers** (Lyu & Li, 2019) and **(2)** an **empirical scaling law** that links training and test loss. When the train set is *non-separable* for all optimized networks, test loss typically tracks training loss, so the latter predicts generalization. Once the model fits the train set (*separability*), margin maximization induces monotonic norm growth, making smaller norms indicative of better generalization. Twin exploits this structure by detecting separability via training accuracy and switching between two validation-free predictors: training loss and parameters' norms.

Practically, Twin performs a logarithmic grid search over LR and WD, supporting early-stopping schedulers to reduce computational burden. The resulting pipeline is simple, model-agnostic, and eliminates the need for a two-step tuning pipeline or saves validation data collection costs when searching for optimal LR and WD. On 37 dataset-architecture configurations, including convolutional (ConvNet), vision transformer (ViT), and feedforward (FF) networks, Twin achieves a mean absolute error of just 1.28% compared to an *Oracle* pipeline that selects HPs based on test accuracy. In summary, the contributions of our paper are as follows:

1. We introduce a validation-free generalization predictor linking margin maximization for homogeneous classifiers (Lyu & Li, 2019) to an empirical train-test loss scaling law, which reveals when training loss or parameters' norm provides monotonic signals of generalization.

2. We ground the analytical findings in Twin, a simple and novel HP-selection pipeline that practically optimizes LR and WD without validation sets, avoiding a cumbersome two-step tuning process.

3. We demonstrate Twin's practical benefits by achieving better performance in small-data regimes, where validation data is scarce, and by reducing validation data collection in medical imaging.

4. We conduct a comprehensive empirical study, spanning diverse datasets, architectures, and data scales, complemented by analyses that showcase the robustness of Twin across experimental setups.

## 2 Problem Definition

We consider an *HP optimization (HPO) problem*, formalized as a tuple $\mathrm{HPO} = (\mathrm{A}, \mathrm{Q}, \mathrm{S})$, where:

1. A is a *learning algorithm*, i.e., a stochastic function mapping data, HPs, and budget to an hypothesis

$$\mathrm{A} : \mathcal{D} \times \mathcal{A} \times \Lambda \times T \to \mathcal{H}. \tag{1}$$

Here $\mathcal{D} := \mathcal{X} \times \mathcal{Y}$ is the data space, $\mathcal{A}$ and $\Lambda$ respectively denote the HP spaces we focus on in this work, i.e., LR and WD, $T$ is a computational budget in terms of learning iterations, and $\mathcal{H}$ the hypothesis space.

2. Q is a *sampler* that proposes HP configurations. Formally, it may depend on previously trained hypotheses, datasets, and metric evaluations:

$$\mathrm{Q} : (\mathcal{H}) \times (\mathcal{D}) \times (\mathcal{M}) \to \mathcal{A} \times \Lambda. \tag{2}$$

The arguments in parentheses are optional: static strategies such as grid search, ignore history, whereas more complex adaptive ones (e.g., Bayesian optimization) condition on past hypotheses and their fitness. $\mathcal{M}$ is a performance metric mapping hypotheses and data to a scalar score, more formally $\mathcal{M} : \mathcal{H} \times \mathcal{D} \rightarrow \mathbb{R}$.

3. S is a *scheduler* that allocates computational resources across candidate hypotheses. It may assign the full budget to each trial or terminate unpromising runs early, depending on partial metric evaluations:

$$\text{S} : (\mathcal{H}) \times (\mathcal{D}) \times (\mathcal{M}) \rightarrow T. \tag{3}$$

Given these three components, the HPO process produces a set of $N$ candidate hypotheses

$$\mathcal{H}_{\text{HPO}} = \left\{ h_i = \text{A}(\mathcal{D}_{\text{train}}, \alpha_i, \lambda_i, \tau_i) \mid (\alpha_i, \lambda_i) = \text{Q}(\cdot), \ \tau_i = \text{S}(\cdot), \ i = 1, \dots, N \right\}, \tag{4}$$

with the goal of identifying the optimal configuration $(\alpha^\star, \lambda^\star) \in \mathcal{A} \times \Lambda$ yielding the resulting hypothesis $h^\star$ which is expected to maximize the performance $\mathcal{M}$ for the chosen task on test data.

**Learning algorithm and validation.** Let $\mathcal{D}_{\text{train}} = \{(x_i, y_i)\} \subseteq \mathcal{D}$ and $\mathcal{D}_{\text{test}} = \{(x_j, y_j)\} \subseteq \mathcal{D}$ be i.i.d. samples from $P(X, Y)$. The hypotheses are actually implemented via a deep network $f(\cdot, \theta) : \mathcal{X} \rightarrow \mathcal{Y}$ with $\theta \in \mathbb{R}^d$, i.e., $h(\cdot) = f(\cdot, \theta)$. The canonical instantiation of the metric $\mathcal{M}$ is the empirical risk $\bar{\mathcal{L}}$ and, together with $\mathcal{D}_{\text{train}}$, the WD $\lambda$, and (cross-entropy) loss fuction $\ell$, defines the regularized training objective

$$\widehat{\mathcal{L}}(\theta) = \bar{\mathcal{L}}(f(\cdot, \theta), \mathcal{D}_{\text{train}}) + \tfrac{\lambda}{2}\|\theta\|^2, \quad \text{with} \quad \bar{\mathcal{L}}(h, \mathcal{D}) = \tfrac{\mathcal{L}(h, \mathcal{D})}{|\mathcal{D}|} \quad \text{and} \quad \mathcal{L}(h, \mathcal{D}) = \sum_{(x,y) \in \mathcal{D}} \ell(h(x), y).$$

The parameters $\theta$ are updated following the general gradient-based formula $\theta_{t+1} = \text{U}(\theta_t, \nabla_\theta \widehat{\mathcal{L}}(\theta_t), \alpha_t, \lambda, \phi)$, with $\phi$ representing optimizer states or additional HPs, and $t \leq T$ denoting the iteration index.

Let $\mathcal{D}_{\text{val}} = \{(x_k, y_k)\} \subseteq \mathcal{D}$. In common practice, one sets $\mathcal{D} = \mathcal{D}_{\text{val}}$ and selects HPs by solving the objective

$$h_{\text{v}}^\star = \arg\min_{h \in \mathcal{H}_{\text{HPO}}} \bar{\mathcal{L}}(h, \mathcal{D}_{\text{val}}). \tag{5}$$

This validation-based procedure serves as a surrogate for the population risk $\mathbb{E}_{(x,y) \sim P(X,Y)}\left[\ell(h(x), y)\right]$ but suffers from sampling noise for validation size $m$ (deviation $\mathcal{O}(1/\sqrt{m})$, Appendix A) and typically requires either holding out a sufficiently large validation set or performing a two-step training pipeline (Figure 1).

**Scope of our work: validation-free HP selection.** We focus on the case where no held-out validation set is available or desirable. Instead of $\bar{\mathcal{L}}(h, \mathcal{D}_{\text{val}})$, we aim to design a *surrogate generalization score* computed solely as a function of training-time information, i.e., $\mathcal{M} = \mathcal{G}(h, \mathcal{D}_{\text{train}})$, where $\mathcal{G}$ is a function possibly composed of several modules. Therefore, the model selection objective we are aiming to solve is

$$h_{\text{vf}}^\star = \arg\min_{h \in \mathcal{H}_{\text{HPO}}} \mathcal{G}(h, \mathcal{D}_{\text{train}}), \tag{6}$$

provided that the function $\mathcal{G}$ can reliably rank candidate hypotheses without recourse to held-out validation data. More formally, our HPO process should yield a hypothesis $h_{\text{vf}}^\star$ such that $\bar{\mathcal{L}}(h_{\text{vf}}^\star, \mathcal{D}_{\text{test}}) \leq \bar{\mathcal{L}}(h_{\text{v}}^\star, \mathcal{D}_{\text{test}})$.

**Relation to other research directions.** It is essential to situate our contribution in relation to other, potentially distinct lines of research. We give a technical comparison here and defer a broader literature review to Section 5. First, we solve a different problem from *validation-free early-stopping* approaches (e.g., Forouzesh & Thiran, 2021) that, according to our notation, design a particular class of schedulers S. Specifically, they optimize $t^\star = \arg\min_{1 \leq t \leq T} \mathcal{G}(h_t, \mathcal{D}_{\text{train}})$ which shares with Equation (6) the *surrogate generalization score* $\mathcal{G}$ but misses explicit design to serve as a valid cross-trial ranking metric making a direct comparison impossible. Second, we partially differ from *generalization-gap prediction* approaches, (e.g., Jiang et al., 2018), which estimate the difference $\Delta(h) = \bar{\mathcal{L}}(h, \mathcal{D}_{\text{train}}) - \bar{\mathcal{L}}(h, \mathcal{D}_{\text{test}})$ between training and test risk for a given hypothesis. These methods typically *learn* a predictor $\widehat{\Delta}(h)$ from training signals, then rank models by $\mathcal{G}(h, \mathcal{D}_{\text{train}}) = \bar{\mathcal{L}}(h, \mathcal{D}_{\text{train}}) - \widehat{\Delta}(h)$. While sharing our goal of validation-free ranking, such predictors $\widehat{\Delta}(h)$ rely on supervised meta-training and must be carefully calibrated across heterogeneous

HP draws before they yield reliable cross-trial and cross-task rankings, whereas our surrogates $\mathcal{G}(h, \mathcal{D}_{\text{train}})$ are designed to be directly comparable within the candidate pool $\mathcal{H}_{\text{HPO}}$ and do not require expensive meta-training, making a direct comparison unfair. Finally, we note that many prior HPO pipelines may differ in their choice of sampler Q or scheduler S (Falkner et al., 2018; Franceschi et al., 2024). These components are largely orthogonal to our contribution: any alternative sampler or scheduler could in principle be embedded within Twin provided that the designed $\mathcal{G}$ yields hypothesis $h_{\text{vf}}^{\star}$ such that $\bar{\mathcal{L}}(h_{\text{vf}}^{\star}, \mathcal{D}_{\text{test}}) \leq \bar{\mathcal{L}}(h_{\text{v}}^{\star}, \mathcal{D}_{\text{test}})$. As such, differences in Q or S alone do not provide a direct comparison to our method. Our approach is more similar in scope and objective to a broad class of methods that analyze *sharpness* of the loss landscape around a solution, motivated by its correlation with generalization (Andriushchenko et al., 2023). *Sharpness* can indeed be casted as a *surrogate generalization score* $\mathcal{G}$ which is applicable to the hypotheses pool $\mathcal{H}_{\text{HPO}}$.

## 3 Tune without Validation

### 3.1 Preliminaries: Margin maximization in homogeneous classifiers

Our methodology, which aims at selecting LR and WD without validation sets, leverages the theoretical result of margin maximization for homogeneous classifiers from (Lyu & Li, 2019). To provide an overview, we recall the main result we will build upon and refer the interested reader to the original work for additional details. The authors rigorously establish that, in homogeneous models, optimizing the logistic or cross-entropy loss with gradient descent or gradient flow leads to a monotonic increase of a smoothed normalized margin, as long as the training loss drops below a specified threshold. The threshold imposed on $\mathcal{L}_{\text{train}}$ is sufficiently small to guarantee separability of the training data, meaning that the network achieves 100% training accuracy.

More formally, the network output is the logit vector $z = \big(z_1(\theta, x), \ldots, z_C(\theta, x)\big) = f(x, \theta) = h(x) \in \mathbb{R}^C$, where $z_j(\theta, x)$ denotes the logit assigned to class $j$. Given the training set $\mathcal{D}_{\text{train}} = \{(x_i, y_i)\}_{i=1}^{N}$, the loss is the cross-entropy objective summed over all training samples $\mathcal{L}(h, \mathcal{D}_{\text{train}}) = \mathcal{L}(\theta, \mathcal{D}_{\text{train}}) = \sum_{i=1}^{N} -\log(\exp(z_{y_i}(\theta, x_i)) / \sum_{j=1}^{C} \exp(z_j(\theta, x_i)))$. For each training example $(x_i, y_i)$, the *margin* is defined as $q_i(\theta) = z_{y_i}(\theta, x_i) - \max_{j \neq y_i} z_j(\theta, x_i)$, and the margin over the entire dataset is defined to be $q_{\min}(\theta) = \min_{i \in [N]} q_i(\theta)$. A central component of the result in (Lyu & Li, 2019) is the *normalized margin*, whose construction relies on the homogeneity of the network $f$, that is, $f(x, b\theta) = b^k f(x, \theta)$ for all $b > 0$, with $k > 0$ denoting the degree of homogeneity (Li & Arora, 2019; Li et al., 2022). To simplify notation, let $\rho = \|\theta\|_2$ and $\hat{\theta} = \theta / \rho$. For a $k$-homogeneous network, the margin scales with $\rho^k$, making the normalized margin:

$$\bar{\gamma}(\theta) = q_{\min}(\hat{\theta}) = \frac{q_{\min}(\theta)}{\rho^k}. \tag{7}$$

The non-smooth minimum margin $q_{\min}(\theta)$ can be approximated by replacing the min operation with a smooth LogSumExp (LSE) over the sample-wise margins $q_i$. This LSE approximation effectively provides a differentiable surrogate that enables explicit characterization of the smoothed margin as a function of the training loss. Under gradient descent (and similarly under gradient flow) on smooth homogeneous networks trained with cross-entropy, the smoothed normalized $\tilde{\gamma}$ margin is defined as

$$\tilde{\gamma}(\theta) = -\frac{\log\big(e^{\mathcal{L}(\theta, \mathcal{D}_{\text{train}})} - 1\big)}{\rho^k}. \tag{8}$$

This surrogate admits the bound $\tilde{\gamma}(\theta) \leq \bar{\gamma}(\theta) \leq \tilde{\gamma}(\theta) + \frac{\log N}{\rho^k}$ establishing that $\tilde{\gamma}$ closely tracks the true normalized margin $\bar{\gamma}$. Once $\mathcal{L}(\theta, \mathcal{D}_{\text{train}})$ drops below $\tau_{\mathcal{L}} = \log 2$, or equivalently the accuracy reaching $\tau_{\text{acc}} = 100\%$, indicating perfect separability, additional theoretical guarantees apply: $\tilde{\gamma}(\theta)$ becomes provably non-decreasing with further training, and the gap $|\tilde{\gamma}(\theta) - \bar{\gamma}(\theta)| \to 0$ as training continues (Lyu & Li, 2019). The primary practical implication of this theoretical result, as originally suggested by the authors, is that training for a longer period can enlarge the normalized margin and, consequently, may have potential benefits in improving model robustness (Lyu & Li, 2019). Next, we delve deeper into our perspective and how we leverage this result from a different viewpoint, aiming to better predict the generalization of neural network configurations without employing validation sets.

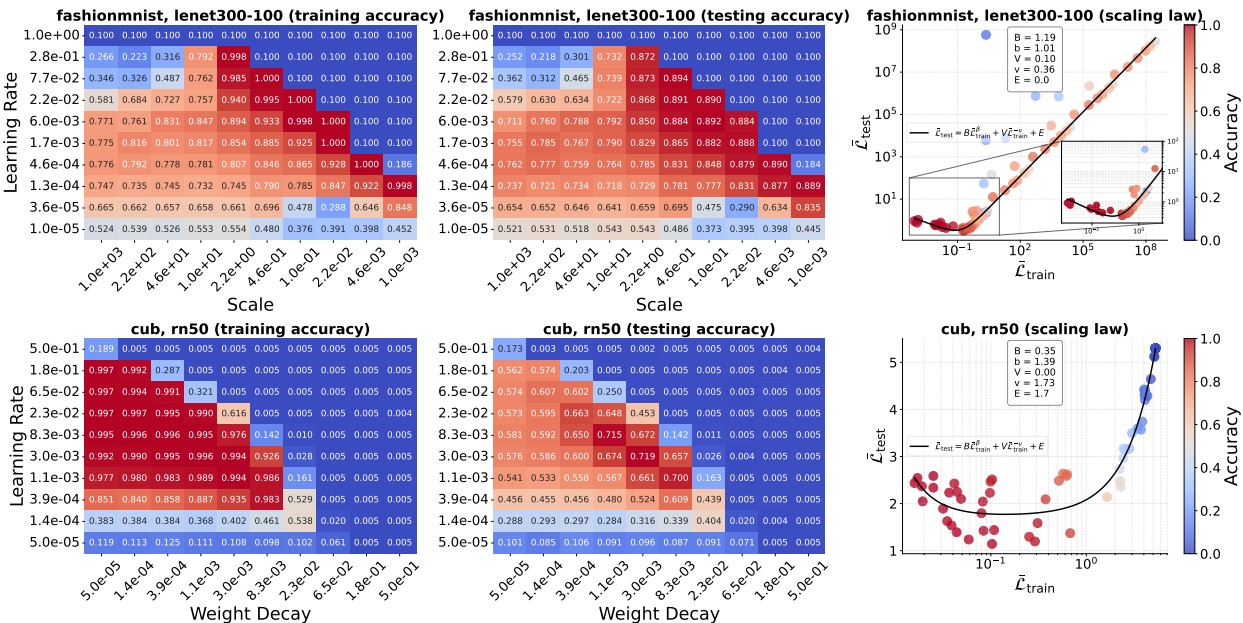

Figure 2: **Optimization regimes and train-test loss scaling laws.** Comparison between the controlled Goldilocks-zone experiment of Vysogorets et al. (2024) (**top row**) and one of our HPO configurations (**bottom row**). Left and middle panels show training and testing accuracy heatmaps, respectively, as functions of the swept HPs (LR and initialization scale vs LR and WD). Both setups exhibit similar staircase-like phase transitions, indicating sharp boundaries between optimization regimes. Right panels report the corresponding train-test loss scaling laws, where each point represents a HP configuration and is color-coded by training accuracy. In both cases, the relation between $\bar{\mathcal{L}}_{\text{test}}$ and $\bar{\mathcal{L}}_{\text{train}}$ follows the characteristic U-shaped curve captured by our Equation (9), suggesting that the scaling law observed within our HPO setups may reflect how optimization dynamics organize homogeneous classifiers across regimes.

## 3.2 Leveraging train-test loss scaling laws and parameters' norms for validation-free HP selection

**Train-test loss scaling law.** We first recall that the HPO problem returns a pool of candidate hypotheses $\mathcal{H}_{\text{HPO}}$ where each $h_i \in \mathcal{H}_{\text{HPO}}$ represents a network $f(\cdot, \theta_i)$ trained up to a budget $T$ of iterations. Let $\mathcal{L}_{\text{HPO}}(\mathcal{D}) = \{\mathcal{L}_i = \mathcal{L}(\theta_i, \mathcal{D}) = \mathcal{L}(h_i, \mathcal{D}) \mid h_i \in \mathcal{H}_{\text{HPO}}\}$ be the formulation referring to the set of loss evaluations for hypotheses performed either on the training ($\mathcal{L}_{\text{HPO}}(\mathcal{D}_{\text{train}})$) or testing ($\mathcal{L}_{\text{HPO}}(\mathcal{D}_{\text{test}})$) sets.

Based on empirical observations across our HPO runs, we hypothesize that train and test losses are related through an underlying scaling law $\mathcal{L}_{\text{test}} = \xi(\mathcal{L}_{\text{train}})$ linking the training loss $\mathcal{L}_{\text{train}}$ to the testing loss $\mathcal{L}_{\text{test}}$. The recurring structure observed across our HPO settings suggests that the relation may capture underlying regime-dependent optimization dynamics. This view is consistent with recent work on the *Goldilocks zone*, which connects trainability to excess positive curvature in the loss landscape and shows how this favorable curvature disappears at hyperparameter extremes (Vysogorets et al., 2024). Precisely, the local geometry of the loss induces distinct learning regimes such as *lazy learning* (Chizat et al., 2019; Kumar et al., 2023), *rich-feature learning* (Kumar et al., 2023), and *divergence* (Liu et al., 2022). Later in the section, we empirically support this connection by comparing the setup of Vysogorets et al. (2024) with one of our HPO settings.

In practice, we specify a compact parametric model that aligns well with empirical scaling-law observations from our HPO setups and is simple enough to admit an explicit derivative analysis, which we leverage to derive generalization metrics that depend only on quantities available at training time. We model the test loss as a sum of three contributions: **(1)** a decaying term associated with the *divergence* regime, **(2)** a growing term linked to *lazy learning*, and **(3)** a floor. We emphasize that this relation should be interpreted as a phenomenological assumption inferred from empirical observations across several training setups, rather than

as a universally established relationship that necessarily holds in all cases. Under this modeling assumption, we consider the following noise-free but analytically convenient parametric form:

$$\mathcal{L}_{\text{test}} = \xi(\mathcal{L}_{\text{train}}) = \underbrace{B\,\mathcal{L}_{\text{train}}^{\beta}}_{\text{decreasing}} + \underbrace{V\,\mathcal{L}_{\text{train}}^{-v}}_{\text{increasing}} + \underbrace{E}_{\text{irreducible floor}}, \tag{9}$$

where $B, V \geq 0$ denote non-negative scaling coefficients, $\beta, v > 0$ control the strength of the respective power-law dependencies, and $E \geq 0$ represents a fixed irreducible error. The above parameters could be empirically estimated from sets $\mathcal{L}_{\text{HPO}}(\mathcal{D}_{\text{train}})$ and $\mathcal{L}_{\text{HPO}}(\mathcal{D}_{\text{test}})$ if $\mathcal{L}_{\text{HPO}}(\mathcal{D}_{\text{test}})$ was available. Equation (9)'s parametric form is also consistent with recent empirical studies reporting that train and test losses are often coupled through (shifted) power laws across architectures and datasets (Brandfonbrener et al., 2025; Mayilvahanan et al., 2025). In practice, real train-test loss scaling curves exhibit fluctuations as models begin to fit their training set. For clarity of exposition, we omit the noise term and discuss its exclusion next.

**Comparison with Vysogorets et al. (2024) optimization regimes.** The interpretation above is supported by recent theoretical work on optimization regimes in deep homogeneous classifiers. Fort & Scherlis (2019) discovered that a large excess of positive curvature and local convexity of the loss Hessian is associated with highly trainable initial points (i.e., parameters' norms) located in this *Goldilocks zone*. Building on this, Vysogorets et al. (2024) provide a mechanistic explanation, showing that such favorable curvature disappears at hyperparameter extremes: overconfident logits induce softmax saturation and vanishing gradients (*lazy learning*), while uniform softmax outputs lead to vanishing gradients (*divergence*). Between these extremes lies the *Goldilocks zone*, where curvature is well-conditioned and both optimization and generalization are favorable. Importantly, these regimes can be traversed not only via initialization scale as originally thought by Fort & Scherlis (2019), but also through other transformations such as softmax temperature scaling, which directly modulate curvature in homogeneous networks (Vysogorets et al., 2024).

To empirically connect these findings to our setting, we replicate the experimental protocol of Vysogorets et al. (2024) (Section 5), training a homogeneous feedforward LeNet300-100 on FashionMNIST with full-batch gradient descent while sweeping LR and initialization scale. We then compare this setup to one of our HPO configurations (ResNet50 on CUB) where we sweep LR and WD. As shown in Figure 2, both setups exhibit highly similar staircase-like phase transitions in training and testing accuracy, reflecting the sharp regime boundaries predicted by Vysogorets et al. (2024). Moreover, when plotting $\bar{\mathcal{L}}_{\text{test}}$ against $\bar{\mathcal{L}}_{\text{train}}$, the replicated experiment produces the same characteristic U-shaped relation of our Equation (9). Beyond this controlled comparison, we provide empirical validation across our 37 dataset–model configurations in Appendix B. These results suggest that the scaling law we observe within our HPO setup may reflect how optimization dynamics organize homogeneous classifiers across regimes.

**Regime transition.** First, to analyze the transition between different regimes, we compute the derivative of the test loss $\mathcal{L}_{\text{test}}$ with respect to the training loss $\mathcal{L}_{\text{train}}$:

$$\frac{d\mathcal{L}_{\text{test}}}{d\mathcal{L}_{\text{train}}} = B\beta\mathcal{L}_{\text{train}}^{\beta-1} - Vv\mathcal{L}_{\text{train}}^{-(v+1)}. \tag{10}$$

Setting the derivative to zero identifies the critical point $C$ corresponding to the minimum of the test loss:

$$\frac{d\mathcal{L}_{\text{test}}}{d\mathcal{L}_{\text{train}}} = 0 \quad \Longleftrightarrow \quad \mathcal{L}_{\text{train}} = \left(\frac{Vv}{B\beta}\right)^{\frac{1}{\beta+v}} = C. \tag{11}$$

In parallel, by rearranging Equation (8), we can express the training loss as a deterministic function of the parameters' norm $\rho$ and smoothed normalized margin $\tilde{\gamma}$. For convenience, we hence derive

$$\mathcal{L}_{\text{train}} = \text{logistic}\left(\rho^k\tilde{\gamma}\right) = \log\left(1 + e^{-\rho^k\tilde{\gamma}}\right). \tag{12}$$

Now, using Equation (12), we express the critical condition in terms of the smoothed normalized margin $\tilde{\gamma}$:

$$\mathcal{L}_{\text{train}} = \log(1 + e^{-\rho^k \tilde{\gamma}}) = C \quad \implies \quad \tilde{\gamma}_C = -\frac{\log(e^C - 1)}{\rho_C^k}, \tag{13}$$

where $\tilde{\gamma}_C$ and $\rho_C^k$ respectively correspond to the smoothed normalized margin and the norm at the critical point $C$. This relation makes explicit that separability ($\tilde{\gamma} = 0$, equivalently $\mathcal{L}_{\text{train}} = \log 2$) acts as a natural reference point for the location of the stationary points of the train–test scaling law. In particular, separability roughly marks the boundary at which the increasing term begin to influence the test loss. When $C \leq \log 2$, the stationary point lies in the separable regime ($\tilde{\gamma}_C \geq 0$). Instead, when $C > \log 2$, the stationary point is necessarily non-separable ($\log(e^C - 1) > 0 \implies \tilde{\gamma}_C < 0$). The distance from the separability boundary in margin space is dampened by $\rho_C^k$, which is usually large in deep networks, making $\tilde{\gamma}_C \approx 0$ in practice.

As a consequence, if none of the models in $\mathcal{L}_{\text{HPO}}(\mathcal{D}_{\text{train}})$ reaches separability, then either no stationary point is present at all, or any stationary point that may arise lies in a neighborhood of the separability boundary whose size is controlled by the network norm, making the contribution of the increasing term negligible in practice. This observation hence justifies using the train loss as a generalization predictor when $\mathcal{L}_{\text{test}} \approx B\mathcal{L}_{\text{train}}^{\beta} + E$.

**Norm-based generalization prediction in the separable regime.** When $C \leq \log 2$, i.e., the critical point belongs to the separable regime, it is more likely that the scaling law transitions from low to high test-loss values. In this more interesting case for validation-free generalization prediction, namely, when the increasing term contributes meaningfully to $\mathcal{L}_{\text{test}}$, we will employ the parameters' norm $\rho$ as a predictor. To study this relation, we compute the derivative of the test loss with respect to $\rho$. By leveraging Equation (10) and Equation (12), the chain rule yields:

$$\frac{d\mathcal{L}_{\text{test}}}{d\rho} = \frac{d\mathcal{L}_{\text{test}}}{d\mathcal{L}_{\text{train}}} \cdot \frac{d\mathcal{L}_{\text{train}}}{d\rho} = \left(B\beta\mathcal{L}_{\text{train}}^{\beta-1} - Vv\mathcal{L}_{\text{train}}^{-(v+1)}\right) \cdot \left(-\frac{k\,\rho^{k-1}\tilde{\gamma}}{1 + e^{\rho^k\tilde{\gamma}}}\right). \tag{14}$$

From the previous analysis, we know that $C < \log 2 \implies \tilde{\gamma} > 0$. Furthermore, being $k > 0$, $\rho > 0$, and the denominator $1 + e^{\rho^k\tilde{\gamma}}$ strictly positive make the entire second factor of Equation (14) negative. Thus, the sign of the derivative $\frac{d\mathcal{L}_{\text{test}}}{d\rho}$ is entirely determined by the negative of the sign of the first factor:

$$\text{sign}\left(\frac{d\mathcal{L}_{\text{test}}}{d\rho}\right) = -\text{sign}\left(B\beta\mathcal{L}_{\text{train}}^{\beta-1} - Vv\mathcal{L}_{\text{train}}^{-(v+1)}\right). \tag{15}$$

In particular, for the common case where the increasing contribution dominates ($Vv\mathcal{L}_{\text{train}}^{-(v+1)} > B\beta\mathcal{L}_{\text{train}}^{\beta-1}$), we get $\frac{d\mathcal{L}_{\text{test}}}{d\rho} > 0$. The derivative indicates that the test loss increases with increasing parameters' norm, hence providing a clear mechanism for using the parameters' norm as a predictor. Conversely, if the decreasing term remains dominant beyond the onset of separability ($B\beta\mathcal{L}_{\text{train}}^{\beta-1} > Vv\mathcal{L}_{\text{train}}^{-(v+1)}$), norm-loss monotonicity can locally be flipped. In this case, the scaling law may admit a stationary point whose location in margin space is confined to a small neighborhood of the separability threshold due to norm scaling for the same argument of the previous paragraph ($\tilde{\gamma}_C \approx 0$). Therefore, the interval in which the decreasing term locally reverses the norm–loss monotonicity is negligible in practice. In summary, for increasing elements, which can again be reliably checked via separability, the parameters' norm $\rho$, being a validation-free signal, is predictive of generalization given the monotonic increasing relation $\frac{d\mathcal{L}_{\text{test}}}{d\rho} \geq 0$ for $\mathcal{L}_{\text{train}} \in (0, C]$.

**Visual illustration from experimental setup.** To complement the previous analysis and provide additional insights, we select five dataset-model pairs from our experimental setup (Section 4) that cover different scaling laws. In Figure 3, we show the train-test loss scaling laws (top row) and test loss vs parameter norm $\rho$ (bottom row). Due to the popularity of the average losses (rather than their sum), we fit Equation (9) on $\bar{\mathcal{L}}_{\text{HPO}}(\mathcal{D}_{\text{train}})$ and $\bar{\mathcal{L}}_{\text{HPO}}(\mathcal{D}_{\text{test}})$. Each dot represents a model trained with an LR-WD couple, and both

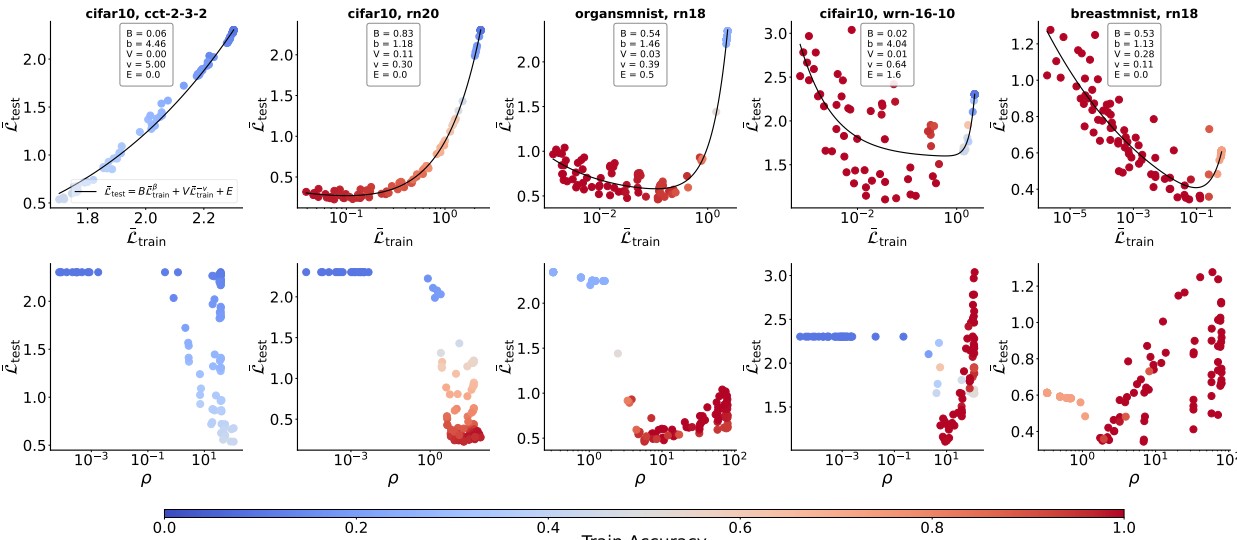

Figure 3: **Train-test loss scaling laws and norm dynamics.** Five dataset–model pairs illustrating how the fitted scaling law (**top**) and test loss vs. norm relation (**bottom**) evolve across learning regimes. Each point corresponds to an LR–WD configuration, color-coded by training accuracy. The leftmost example has no critical point and the train loss is predictive of test loss. The remaining columns show that the parameters' norm $\rho$ becomes predictive of generalization within **fully separable** settings.

parameters are swept over logarithmic intervals (see next for more details). The color coding, instead, high-lights the training accuracy, i.e., the *separability* indicator. In a decreasing-term-dominated regime (first column), there is no critical point $C$, and the training loss is indeed predictive of generalization. In contrast, the remaining columns from left to right progressively illustrate more increasing-term-dominated settings, ranging from cases where only a subset of configurations achieves high training accuracy to fully separable regimes where nearly all models attain 100% accuracy. Once separability is (approximately) reached, the fitted scaling law may exhibit the characteristic bend induced by the increasing term $V\bar{\mathcal{L}}_{\text{train}}^{-v}$, causing $\bar{\mathcal{L}}_{\text{test}}$ to increase as $\bar{\mathcal{L}}_{\text{train}}$ decreases. This transition is accompanied by an equally progressive rise in the parameters' norm $\rho$ (bottom row), which becomes a reliable predictor of generalization within the separable regime.

**On the specific focus for LR and WD as HPs.** Although the proposed scaling-law analysis is, in principle, agnostic to the choice of HPs, we concentrate on LR and WD for both theoretical and practical reasons. In our setting, the training loss $\mathcal{L}_{\text{train}}$ is directly and sensitively modulated by the LR, making it a primary driver of the optimization regime in which the model operates. Conversely, the parameters' norm $\rho$ is strongly influenced by WD, which regulates norm growth and thereby shapes the trajectory of the normalized margin and the onset of the increasing term in Equation (9). Together, LR and WD provide complementary controls over the optimization dynamics, allowing the HPO procedure to traverse different regimes and consistently exhibit the observed scaling-law behavior across our experiments.

At the same time, prior work shows that similar optimization-induced regime transitions can arise through other mechanisms such as changes in initialization scale or softmax temperature (Vysogorets et al., 2024). This suggests that the behavior captured by our scaling law is not specific to LR and WD, but rather reflects more general properties of the optimization dynamics.

Therefore, while LR and WD represent popular HPs to optimize as done in our experiments, other reg-ularization or optimization strategies that influence norm growth or loss curvature could, in principle, be incorporated into the same framework. To empirically validate this hypothesis, we applied Twin to the distinctly different setup of Vysogorets et al. (2024), which sweeps the initialization scale rather than WD of a LeNet300-100 on FashionMNIST (Figure 2). Twin successfully identifies a near-optimal configuration,

achieving 89.18% test accuracy compared to the maximum 89.41%. Beyond initialization scale, Balestriero et al. (2022) show that data augmentation can systematically affect parameters' norms, indicating that additional HPs may be amenable to similar treatment. This opens the door to future extensions of our approach, provided their effect on the underlying optimization regime can be characterized with sufficient consistency.

**Practical approximations.** Finally, in our analysis, we made some simplifying assumptions concerning the scaling law, the smoothed margin, and exact $k$-homogeneity, which we discuss further in Appendix C. Despite these approximations, the qualitative predicted behavior remains remarkably stable and effective in practice (Section 4), providing a validation-free mechanism for identifying suitable LR and WD.

### 3.3 Practical tuning pipeline

We now translate the theoretical insights of the previous section into a practical, validation-free HPO procedure. The resulting Twin pipeline (Algorithm 1) implements a simple rule informed by the modeling of Section 3.2: **(1)** If any configuration reaches the separable regime (training-accuracy threshold $\tau_{\mathrm{acc}}$), then the model with the smallest parameters' norm is selected; **(2)** Otherwise, the configuration with the lowest training loss is chosen. This decision rule mirrors the structure revealed by our analysis: train loss is predictive of generalization in the *divergence* regime, while parameters' norms are predictive in the *lazy learning* regime. Twin consists of a sampler Q, a scheduler S, and a validation-free surrogate metric $\mathcal{M}$ that implements the separability-informed selection rule.

---

**Algorithm 1** Twin

**Require:** LR range $[\alpha_{\min}, \alpha_{\max}]$, #LRs $N_\alpha$, WD range $[\lambda_{\min}, \lambda_{\max}]$, #WDs $N_\lambda$, separability threshold $\tau_{\mathrm{acc}}$, scheduler $S \in \{\mathrm{FIFO}, \mathrm{HB}_{25\%}, \mathrm{HB}_{12\%}\}$
**Ensure:** Optimal configuration $(\alpha^\star, \lambda^\star)$
  # Search with trial scheduling
1: $\Psi, \Theta, \mathcal{P} \leftarrow \texttt{zeros}(N_\alpha, N_\lambda)$
2: $\{\alpha_i\} \leftarrow \texttt{logsweep}(\alpha_{\min}, \alpha_{\max}, N_\alpha)$
3: $\{\lambda_j\} \leftarrow \texttt{logsweep}(\lambda_{\min}, \lambda_{\max}, N_\lambda)$
4: **for** $i = 1, \ldots, N_\alpha$ **do**
5:     **for** $j = 1, \ldots, N_\lambda$ **do**
6:        $f_\theta \leftarrow \texttt{runtrial}(\mathcal{D}_{train}, \alpha_i, \lambda_j, S)$
7:        $\Psi[i, j] \leftarrow \bar{\mathcal{L}}(f_\theta, \mathcal{D}_{train})$    # Store average train loss
8:        $\Theta[i, j] \leftarrow \|\theta\|$         # Store parameters' norm
9:        $\mathcal{P}[i, j] \leftarrow \mathrm{Accuracy}(f_\theta, \mathcal{D}_{train})$   # Store train accuracy
10:    **end for**
11: **end for**
  # Validation-free HP selection
12: $\mu \leftarrow \mathcal{P} > \tau_{\mathrm{acc}}$        # Mask for separable configs
13: **if** $\texttt{any}(\mu)$ **then**       # Check for separable regime
14:    $(\alpha^\star, \lambda^\star) \leftarrow \texttt{argmin}(\Theta[\mu])$
15: **else**
16:    $(\alpha^\star, \lambda^\star) \leftarrow \texttt{argmin}(\Psi)$
17: **end if**

---

**Sampler Q: grid search.** Twin executes a total of $N = N_\alpha \cdot N_\lambda$ training trials, where $N_\alpha$ and $N_\lambda$ denote the number of LR and WD values, respectively. By default, Q is instantiated as a logarithmic grid search: the `logsweep` function generates values over the ranges $[\alpha_{\min}, \alpha_{\max}]$ and $[\lambda_{\min}, \lambda_{\max}]$ on a log scale. The resulting sets $\{\alpha_i\}_{i=1}^{N_\alpha}$ and $\{\lambda_j\}_{j=1}^{N_\lambda}$ define the HP configurations explored during the search. Although grid search is less efficient than alternatives (e.g., random search), it remains a common choice for tuning LR and WD (Kornblith et al., 2019; Steiner et al., 2022). Furthermore, Twin remains competitive even with coarse and sparse grid resolutions (Section 4.3). To lower the computational cost, Twin integrates early-stopping schedulers, as we describe in more detail below.

**Scheduler S: FIFO or adapted HyperBand.** We consider FIFO (no early stopping) and a modified HyperBand algorithm in which the standard successive-halving procedure runs until only an $X\%$ subset of promising configurations remains active that then train to completion. This modification stabilizes the loss landscape, avoiding premature pruning of LR–WD configurations that might otherwise achieve separability. We denote this variant $\mathrm{HB}_{X\%}$ and evaluate $X \in \{25\%, 12\%\}$ in Section 4.3. Practitioners can safely default to $\mathrm{HB}_{25\%}$. Since Twin is validation-free, the scheduler evaluates the goodness of trials considering the average train loss over five epochs.

**Validation-free metric $\mathcal{M}$: separability-informed decision rule.** The final stage replaces a held-out validation set with a surrogate metric $\mathcal{M} = \mathcal{G}(h, \mathcal{D}_{\mathrm{train}})$ directly derived from the analysis of Section 3.2. Concretely, the grid search produces three log-structured matrices over the LR–WD space: the matrix of average training losses, $\Psi \in \mathbb{R}^{N_\alpha \times N_\lambda}$, where $\Psi[i, j]$ is the final empirical risk $\bar{\mathcal{L}}(f_\theta, \mathcal{D}_{\mathrm{train}})$ for configuration

$(\alpha_i, \lambda_j)$, the matrix of parameters' norms, $\Theta \in \mathbb{R}^{N_\alpha \times N_\lambda}$, with $\Theta[i,j] = \|\theta\|$, and the accuracy matrix $\mathcal{P} \in \mathbb{R}^{N_\alpha \times N_\lambda}$. A single threshold $\tau_{\mathrm{acc}}$ determines separability. Due to the previously mentioned approximations and consequent noise (e.g., smoothed margin), we slightly relax the separability threshold ($\tau_{\mathrm{acc}} = 100\%$) and set it to $\tau_{\mathrm{acc}} = 99\%$ for all experiments. We further elaborate on this choice in Section 4.3. If at least one configuration satisfies $\mathcal{P} > \tau_{\mathrm{acc}}$, the HPO problem is (very likely) in the increasing regime, and the theory predicts that the test loss is monotonic increasing in the parameters' norm. Twin therefore selects the model with the smallest norm among separable configurations by first setting a mask $\mu = \mathcal{P} > \tau_{\mathrm{acc}}$, and then computing $(\alpha^\star, \lambda^\star) = \texttt{argmin}(\Theta[\mu])$. If no configuration reaches separability, the process is in the regime where train loss is decreasing and hence predictive. Twin then selects the model with minimal $\bar{\mathcal{L}}(f_\theta, \mathcal{D}_{\mathrm{train}})$, i.e., $(\alpha^\star, \lambda^\star) = \texttt{argmin}(\Psi)$.

This separability-informed selection rule is simple, fully validation-free, and follows directly from the structure revealed by Equation (9) and smoothed margin evolution of Section 3.2.

# 4 Experiments

We structure our experimental evaluation section as follows. First, in Section 4.1, we compare Twin against sharpness-based selection to highlight the limitations of existing validation-free approaches and motivate our approach. Second, in Section 4.2, having established Twin's efficacy, we conduct a comprehensive evaluation against traditional validation-based selection across three distinct domains: small datasets (Section 4.2.1), medical imagery (Section 4.2.2), and natural images (Section 4.2.3). Finally in Section 4.3, we perform several sensitivity analyses concerning the separability threshold, robustness to grid sparsity, impact of early stopping, and the use of different optimizers and schedulers.

## 4.1 Comparison with Sharpness-Based Selection

**Motivation.** The sharpness of minima has long been studied as a proxy for generalization in deep learning (Hochreiter & Schmidhuber, 1994; Jiang et al., 2020; Andriushchenko et al., 2023). Both average-case and worst-case sharpness have been considered in the literature, with worst-case sharpness generally correlating more strongly with generalization, particularly when evaluated over small perturbation sets and subsets of the full training sample ($m$-sharpness) (Jiang et al., 2020; Dziugaite et al., 2020; Kwon et al., 2021; Foret et al., 2021). This intuition underpins methods like Sharpness-Aware Minimization (SAM) (Foret et al., 2021), where optimizing for flatter minima can improve test accuracy. However, SAM does not explicitly target HP tuning but rather alters the training dynamics to reach more generalizable solutions. As such, comparing Twin directly to SAM is not appropriate, since SAM modifies the training process, while Twin focuses on selecting among trained models the best without accessing the validation set. For this reason, we compare Twin to worst-case $m$-sharpness, a post-hoc measure of the sharpness around the final solution. This remains within the scope of validation-free HP selection, as lower sharpness has been shown to correlate with better generalization (Jiang et al., 2020), yet not in all configurations (Andriushchenko et al., 2023).

**Baseline: worst-case $m$-sharpness.** Let $\mathcal{D}_{\mathrm{s}} = \{(x_k, y_k)\}$ be a subset of the full training set $\mathcal{D}_{\mathrm{train}}$ (i.e., $\mathcal{D}_{\mathrm{s}} \subseteq \mathcal{D}_{\mathrm{train}}$) and $f_\theta$ a neural network with parameters $\theta$, trained to minimize the empirical risk $\bar{\mathcal{L}}$. The worst-case $m$-sharpness of a model $f_\theta$ is defined as

$$s(f_\theta, \mathcal{D}_{\mathrm{s}}) = \mathbb{E}_{\mathcal{D}_m \sim \mathcal{D}_{\mathrm{s}}} \left[ \max_{\|\delta\|_2 \leq \rho_s} \bar{\mathcal{L}}(f_{\theta+\delta}, \mathcal{D}_m) - \bar{\mathcal{L}}(f_\theta, \mathcal{D}_m) \right], \tag{16}$$

where $\delta$ is an adversarial perturbation bounded in $\ell_2$-norm by $\rho_s$ and $\mathcal{D}_m$ represents the sampled batches ($|\mathcal{D}_m| = m$). This quantity estimates the local sharpness of the loss landscape around the final solution $\theta$ without retraining and can be used to rank model generalization. To compute worst-case $m$-sharpness, we mainly follow Andriushchenko et al. (2023). Specifically, we employ an HP-free algorithm (Auto-PGD, Jiang et al., 2018) in the parameter space to solve the maximization problem where perturbations $\delta$ are bounded within an $\ell_2$-ball of radius $\rho_s = 0.05$. We evaluate sharpness on mini-batches of size $m = 128$ out of $|\mathcal{D}_{\mathrm{s}}| = 2048$ total samples, using 20 gradient ascent steps. We do not use gradient normalization or adaptive

updates. We observed that sharpness alone is not a reliable selection proxy across all trials. Specifically, the configurations resulting in exceptionally poor training exhibit extremely low sharpness, likely because the optimizer converges to flat but high-loss regions. To avoid such degenerate solutions, we filter out any configuration achieving less than 15% training accuracy. We chose this threshold as a conservative heuristic floor to exclusively filter out models that suffered from complete training collapse. This ensures that the evaluated sharpness values reflect meaningful properties of well-optimized models, rather than failed training runs. Still, it is worth noting that without this tweak to filter poorly optimized configurations in our experiments, a naive application of $m$-sharpness would select trials that result in indistinguishable from or slightly above random-chance accuracy.

**Datasets.** For this comparison, we select popular natural-image datasets such as CIFAR-10/100 (Krizhevsky, 2009) and Tiny Imagenet (Le & Yang, 2015). These datasets contain 50,000/100,000 training samples from 10 to 200 classes, and image resolutions of $32 \times 32$ and $64 \times 64$.

**Implementation details.** We set the LRs and WDs intervals for the grid search to $[5 \cdot 10^{-5}, 5 \cdot 10^{-1}]$ for ConvNets, and to $[10^{-6}, 10^{-1}]$ for ViTs. We employ the FIFO scheduler for both Twin and $m$-sharpness. On CIFAR, we employ as ConvNets a ResNeXt-11 (4 x 16d) (RNX11) (Xie et al., 2017) and a ResNet of depth 20 (RN20) (He et al., 2016). As ViTs, we train architectures designed explicitly for CIFAR, such as the Compact Convolutional Transformer with two encoder and convolutional-stem layers (CCT-2/3×2) (Hassani et al., 2021). On TinyImagenet, we train a WRN-16-4 (Zagoruyko & Komodakis, 2016) and a Compact Vision Transformer (Hassani et al., 2021) with 7 encoder layers and a patch size of 8 (CVT-7/8).

**Twin outperforms $m$-sharpness on ViTs and is comparable on ConvNets.** Table 1 reports the comparison between Twin and worst-case $m$-sharpness across natural image datasets (CIFAR-10/100 and TinyImagenet) and network architectures (ConvNets and ViTs). We observe that Twin outperforms $m$-sharpness in 6 out of 8 experimental scenarios and achieves a higher average test accuracy (73.4% vs. 71.3%). While sharpness is known to correlate with generalization in some cases (Jiang et al., 2020), our results highlight that this correlation is not always strong enough to reliably guide optimal HP selection. The performance gap is espe-

| Dataset | Model Class | Model | # Trials | $m$-sharpness | Twin | $\Delta$ |
|---|---|---|---|---|---|---|
| C10 | ViT | CCT-2/3×2 | 49 | 82.0 | **87.3** | +5.3 |
| C10 | ConvNet | RN20 | 100 | 91.6 | **91.8** | +0.2 |
| C10 | ConvNet | RNX11 | 100 | 90.0 | **90.2** | +0.2 |
| C100 | ViT | CCT-2/3×2 | 49 | 55.4 | **64.0** | +8.6 |
| C100 | ConvNet | RN20 | 100 | **67.8** | 67.6 | -0.2 |
| C100 | ConvNet | RNX11 | 100 | 67.2 | **67.7** | +0.5 |
| TinyIN | ViT | CVT-7/8 | 36 | 55.9 | **58.0** | +2.1 |
| TinyIN | ConvNet | WRN-16-4 | 49 | **60.8** | 60.8 | +0.0 |
| **Average** | | | | 71.3 | **73.4** | +2.1 |
| **# Wins** | | | | 2 | **7** | |

Table 1: **Twin vs. worst-case $m$-sharpness.** Results report test accuracy (%). Details on the computation of $m$-sharpness are available in Section 4.1.

cially pronounced, up to $\approx$9 percentage points, for ViTs (CCT-2/3×2 and CVT-7/8), where prior work has also observed a weaker correlation between sharpness and generalization (Andriushchenko et al., 2023).

## 4.2 Comparison with Oracle and Validation-Based Selection

**Motivation.** To further assess the effectiveness of Twin, we introduce two additional baselines that are more robust than sharpness-based selection across experimental setups: namely, the *Selection from Validation Set* (SelVS) and the *Oracle*, or unrealistic selection from the test set. As far as the domains for experimentation are concerned, we aim to cover scenarios where Twin's capabilities could be more advantageous. Firstly, our evaluation encompasses small datasets (Section 4.2.1), a setup particularly suitable for Twin, given that traditional pipelines struggle with HP selection due to the limited dimensions of validation sets, as explained in Section 2. Additionally, we explore Twin's applicability in the medical domain (Section 4.2.2), where its ability to mitigate the need for collecting validation sets is particularly valuable, considering the complexities and regulations inherent in healthcare settings. Finally, we further examine Twin in dealing with natural images (Section 4.2.3), as this domain is widely employed for benchmarking.

**Baselines: SelVS and *Oracle*.** The SelVS is the traditional reference point, where HP optimization is conducted exclusively on the validation set. In our experimental setup, the validation set is either subsampled from the training set (small and natural image datasets) or collected externally (medical data). These two

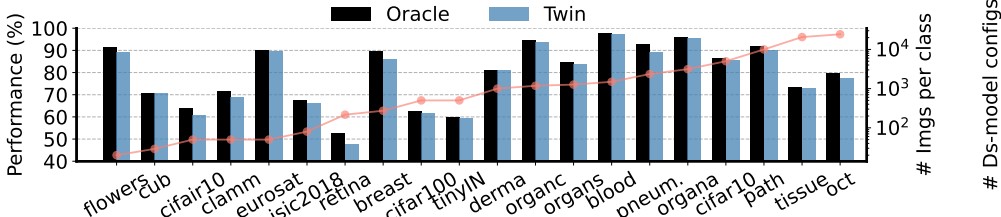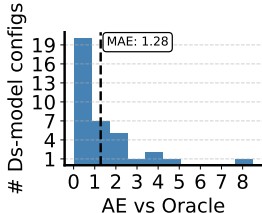

Figure 4: **Quantitative comparison with *Oracle*.** Twin closely matches the *Oracle* on tasks with various training images per class (**left**). Twin scores an overall 1.28% MAE against the *Oracle* pipeline across 37 different dataset-model configurations when using a FIFO scheduler (**right**).

cases correspond to the left two schemes of Figure 1. SelVS can also be performed with different trial schedulers, e.g., with early stopping. The *Oracle* represents the ideal but unrealistic scenario of selecting HPs directly from the test set. The *Oracle* always runs a FIFO scheduler. At its best, Twin achieves 0% mean absolute error (MAE) against the *Oracle*. Both SelVS and *Oracle* select the HPs according to the relevant last-epoch metric with FIFO schedulers and the average of the last five epochs with early stopping.

**Comparison with *Oracle*.** We quantitatively compare Twin against our ideal *Oracle* baseline. In particular, we show the performance per dataset (Figure 4, left) and the absolute errors across 37 different dataset-architecture configurations (Figure 4, right). Twin closely matches the *Oracle* on tasks with various training images per class, scoring an overall 1.28% MAE against it. The significance of this tight alignment is further underscored by the high difficulty of the underlying tuning tasks. As shown in Figure 11, test accuracy is highly sensitive to HP choices, frequently exhibiting high variance (up to 90 percentage points). Consequently, Twin's strong performance is not an artifact of trivial search spaces, but rather demonstrates its robust capability to closely track the *Oracle* accuracy in challenging tuning scenarios.

### 4.2.1 Small Datasets

**Datasets.** For the experiments on small datasets, we select the benchmark introduced by Brigato et al. (2022), which contains five different datasets spanning various domains and data types. In particular, the benchmark contains sub-sampled versions of ciFAIR-10 (Barz & Denzler, 2020), EuroSAT (Helber et al., 2019), CLaMM (Stutzmann, 2016), all with 50 samples per class, and ISIC 2018, with 80 samples per class (Codella et al., 2019). Also, the widely known CUB dataset with 30 images per category is included (Wah et al., 2011). The spanned image domains of this benchmark hence include RGB natural images (ciFAIR-10, CUB), multi-spectral satellite data (EuroSAT), RGB skin medical imagery (ISIC 2018), and grayscale hand-written documents (CLaMM). For EuroSAT, an RGB version is also available. Finally, we include the Oxford Flowers dataset, which comprises 102 categories with 20 images (Nilsback & Zisserman, 2008).

**Implementation details.** Along with the popular ResNet-50 (RN50), which was originally evaluated on the benchmark by Brigato et al. (2022), we also employ EfficientNet-B0 (EN-B0) (Tan & Le, 2019) and ResNeXt-101 (32 × 8d) (RNX101) (Xie et al., 2017) to cover three classes of model scales, respectively *tiny*, *small*, and *base* (Touvron et al., 2021), with 5.3M, 25.6M, and 88.8M parameters. On ciFAIR-10, we employ a Wide ResNet 16-10 (WRN-16-10). We refer the reader to Appendix D.1 for all the details regarding training-related parameters. We perform squared grid searches of 100 trials for RN50 and EN-B0 and 36 trials for RNX101 due to the higher computational burden of the larger RNX101 and to evaluate Twin across grids of different sizes. We set the LRs and WDs intervals for the grid search to $[5 \cdot 10^{-5}, 5 \cdot 10^{-1}]$ to span four orders of magnitude. We report results for Twin and SelVS with early stopping, which respectively employ $HB_{25\%}$ and ASHA as schedulers, with the same number of trials. For the early-stopping parameters, we follow Brigato et al. (2022) and keep a halving rate of two and a grace period of 5% of the total epochs.

**Twin outperforms SelVS by avoiding reliance on small validation sets.** As visible in Table 2, Twin outperforms the traditional SelVS by scoring 69.5% vs 69.2% with EN-B0, 76.2% vs 74% with RN50, and 75.6% vs 73.7% with RNX101 when averaging performance across the CUB, ISIC 2018, EuroSAT, and CLaMM datasets. Indeed, SelVS relies on a small validation set, which may lead to sub-optimal HPs given the lower reliability of the validation error. Furthermore, despite aggressive trial stopping ($HB_{25\%}$), which can make the train-test loss scaling law noisier, Twin still finds robust LR-WD configurations and is thus scalable to computationally intensive search tasks that would be prohibitive without early stopping strategies. When dealing with small datasets, it is also common practice to start with a network pre-trained on a larger

| Method | Model | CUB | ISIC 2018 | EuroSAT | CLaMM | Avg. | # Wins |
|--------|-------|-----|-----------|---------|-------|------|--------|
| *Oracle* | EN-B0 | 67.2 | 66.8 | 91.0 | 65.3 | 72.6 | 4 |
| SelVS | EN-B0 | **67.0** | **65.1** | 86.6 | **58.0** | 69.2 | **3** |
| Twin | EN-B0 | 66.2 | 64.0 | **91.0** | 56.8 | **69.5** | 1 |
| *Oracle* | RN50 | 72.0 | 69.4 | 90.2 | 74.6 | 76.6 | 4 |
| SelVS[†] | RN50 | 70.8 | 64.5 | **90.6** | 70.2 | 74.0 | 1 |
| Twin | RN50 | **72.0** | **68.8** | 89.6 | **74.4** | **76.2** | **3** |
| *Oracle* | RNX101 | 73.0 | 66.7 | 90.0 | 75.2 | 75.9 | 4 |
| SelVS | RNX101 | 72.1 | 62.4 | **90.0** | 70.1 | 73.7 | 1 |
| Twin | RNX101 | **73.0** | **65.8** | 88.6 | **75.2** | **75.6** | **3** |

Table 2: **Small datasets.** The evaluation metric is the balanced test accuracy (%) (Brigato et al., 2022). We allocate 100 and 36 trials for EN-B0/RN50 and RNX101, respectively. Twin and SelVS respectively employ the $HB_{25\%}$ and ASHA schedulers. [†]Values are taken from Brigato et al. (2022).

dataset. Therefore, we also repeat the optimization runs with ImageNet checkpoints. Twin remains effective in transfer learning scenarios, provided that the overlap between the pre-training and target domains is considered. In such cases, adapting the scheduler (e.g., via $HB_{X\%}$) enables Twin to maintain low error, as detailed in Appendix E.

### 4.2.2 Medical Images

**Datasets.** We leverage the MedMNIST v2 benchmark (Yang et al., 2023) to test Twin on medical imaging tasks. We focus on 2D classification and select 11 out of 12 binary/multi-class or ordinal regression tasks of the MedMNIST2D sub-collection, which covers primary data modalities (e.g., X-ray, OCT, Ultrasound, CT, Electron Microscope) and data scales (from 800 to 100,000 samples). The MedMNIST2D benchmark provides held-out validation sets to allow HP tuning. The data diversity of this benchmark presents a significant challenge. We select the testbed with the images pre-processed to $28 \times 28$ resolution out of the full benchmark to maintain the total computational load under a reasonable budget.

**Implementation details.** We use the ResNet-18 (RN18) originally employed in the benchmark by Yang et al. (2023), which consists of four stages but with a modified stem more suitable for low-resolution images. We keep the same Twin configurations tested in Section 4.2.1, except for the trial schedulers that we default to FIFO for both Twin and SelVS. Refer to Appendix D.2 for additional details on the implementation.

**Twin matches SelVS while cutting validation data collection cost.** We summarize the empirical results over MedMNIST2D in Table 3. The *Oracle* scores an upper bound 84.8% test accuracy averaged across the 11 tasks. Twin is comparable to the traditional SelVS (83.1% vs 83.2% and 7 vs 5 wins) and slightly improves its performance in this domain when early stopping is employed (Section 4.3). Twin finds proper HPs and leads to a cost-effective solution by reducing data collection and labeling expenses associated with the ∼10% of samples per dataset originally allocated for validation in the MedMNIST2D benchmark.

### 4.2.3 Natural Images

**Datasets.** As in Section 4.1, we employ the CIFAR-10, CIFAR-100, and Tiny Imagenet benchmarks.

**Implementation details.** On CIFAR datasets, in addition to the models described in Section 4.1, we also consider a WRN-40-2 (Zagoruyko & Komodakis, 2016) and for FF networks, MLPs with batch normalization, ReLU activations, and hidden layers of constant width. Specifically, we use four hidden layers with 256 units on CIFAR-10 and 512 units on CIFAR-100 (MLP-4-256 and MLP-4-512). Data augmentation strength varies from base to medium to strong $\{+, ++, +++\}$. All other training settings and additional details follow the same protocol as outlined in Section 4.1. We provide more details in Appendix D.3.

| Method | Path | Derma | OCT | Pneum. | Retina | Breast | Blood | Tissue | OrganA | OrganC | OrganS | **Avg.** | **# Wins** |
|---|---|---|---|---|---|---|---|---|---|---|---|---|---|
| *Oracle* | 91.9 | 80.8 | 79.8 | 92.8 | 52.5 | 89.7 | 97.8 | 73.2 | 95.9 | 94.5 | 84.4 | 84.8 | 11 |
| SelVS | **90.5** | 80.3 | **78.0** | **92.5** | 46.0 | 85.3 | 96.9 | **72.8** | 94.9 | **94.4** | 83.5 | **83.2** | 5 |
| Twin | 90.0 | **80.8** | 77.3 | 89.3 | **47.8** | **85.9** | **97.1** | **72.8** | **95.3** | 93.7 | **83.8** | 83.1 | **7** |

Table 3: **Medical images.** Twin is more cost-effective by eliminating the need for the held-out validation set (10% of samples are allocated in MedMNIST2D) while being comparable to SelVS. The performance is the test accuracy (%). We run 100 trials per dataset. All approaches employ RN18 and the FIFO scheduler.

**Twin matches SelVS with a single-step tuning pipeline.** Twin is, on average, comparable to SelVS (69.6% vs 69.8%) despite not having access to the validation set (Table 4). Although SelVS achieves more wins, Twin offers similar average performance with a single-step tuning process, avoiding the cumbersome two-step model selection pipeline of first finding the optimal HPs and then training the final model. Remarkably, Twin performs consistently across ConvNets, ViTs, and MLPs, confirming its architecture-agnostic nature, which stems from its reliance on margin-maximization dynamics and homogeneity. Similarly, Twin is agnostic to the strength of data augmentation.

| Dataset | Model Class | Model | # Trials | Aug. | *Oracle* | SelVS | Twin |
|---|---|---|---|---|---|---|---|
| C10 | FF | MLP-4-256 | 100 | + | 66.1 | **65.9** | 65.1 |
| C10 | ViT | CCT-2/3×2 | 49 | +++ | 87.3 | **87.3** | **87.3** |
| C10 | ConvNet | RNX11 | 100 | + | 90.7 | **90.6** | 90.2 |
| C10 | ConvNet | WRN-40-2 | 49 | ++ | 94.0 | 93.3 | **93.7** |
| C100 | FF | MLP-4-512 | 100 | ++ | 35.4 | **35.1** | 34.9 |
| C100 | ViT | CCT-2/3×2 | 49 | +++ | 65.0 | **65.0** | 64.0 |
| C100 | ConvNet | RNX11 | 100 | + | 68.8 | **68.6** | 67.7 |
| C100 | ConvNet | WRN-40-2 | 49 | ++ | 74.2 | 72.8 | **74.2** |
| TinyIN | ViT | CVT-7/8 | 36 | +++ | 58.0 | **58.0** | **58.0** |
| TinyIN | ConvNet | WRN-16-4 | 49 | ++ | 61.8 | **61.8** | 60.8 |
| **Average** | | | | | 70.1 | **69.8** | 69.6 |
| **# Wins** | | | | | 10 | **8** | 4 |

Table 4: **Natural Images.** Although SelVS achieves more wins, Twin offers similar average performance (69.6 vs 69.8) with a single-step tuning process. The reported performance is the test set accuracy (%). The FIFO scheduler is employed. For details on data augmentation (Aug.), refer to Appendix D.3.

### 4.3 Sensitivity Analyses

**Separability threshold.** In this analysis, we focus on the impact of the separability threshold $\tau_{\text{acc}}$, which acts as a practical regime indicator for when variance effects can begin to meaningfully influence the bias–variance scaling law. In Figure 5, we sweep $\tau_{\text{acc}}$ in $[70\%, 80\%, 90\%, 95\%, 99\%, 100\%]$ and measure performance in terms of the MAE against the *Oracle* across 37 different configurations. We observe a clear trend in how the separability criterion affects prediction quality. Increasing the threshold from 70% to 99% steadily improves performance, with the MAE decreasing from 7.8 to 1.3. This reflects the fact that low $\tau_{\text{acc}}$ prematurely classifies many underfitting models as "separable", thereby misidentifying the bias regime and introducing significant errors (norm is not monotonic increasing in $\mathcal{L}_{\text{test}}$). The best performance is achieved at $\tau_{\text{acc}} = 0.99$, which closely matches

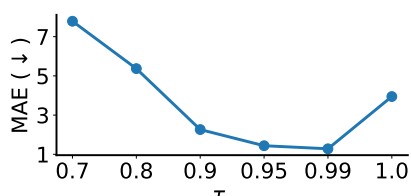

Figure 5: **Separability threshold**. A slight relaxation of the separability threshold enables reliable predictions. The MAE is vs the *Oracle*.

the theoretical separability boundary of 100%. Performance degrades when using the strict 100% accuracy threshold, with MAE rising to 3.9. This behavior is consistent with our theoretical analysis: due to margin smoothing, the surrogate margin $\tilde{\gamma}$ may still be negative even when the true margin $\bar{\gamma}$ is positive, and any stationary point that arises in this regime lies in a small neighborhood of the separability boundary ($\tilde{\gamma}_C \approx 0$), but not necessarily exactly zero. This result confirms that a small relaxation, allowing accuracy just below 100% to count as separable, provides a more robust and numerically stable criterion for predicting optimal LR and WD in practice.

**Grid sparsity.** We further assess the robustness of Twin under increasingly sparse HP sweeps. Instead of using regular log-spaced patterns as before, we randomly remove a fraction of LR–WD configurations and repeat this procedure across 50 seeds. This setting brings the evaluation closer to a random-search regime, while still constrained by the underlying grid, thereby testing Twin under substantially less structured HP ranges. We sweep the percentage of missing grid cells in $[0\%, 20\%, 40\%, 60\%, 80\%]$ and evaluate Twin against the *Oracle* across all 37 configurations. The resulting MAE distributions (Figure 6) remain close to the full-grid setup: the median MAE only increases from 1.3 at 0% removal to 1.4, 1.5, 1.6, and 1.5 at 20%, 40%, 60%, and 80%, respectively. Even with 60%–80% of the grid cells removed, per-

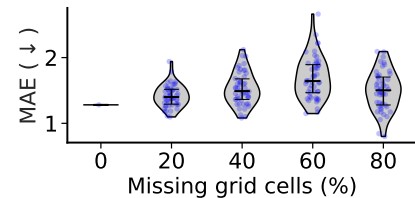

Figure 6: **Grid sparsity.** Twin scores consistent results even with randomly sparse search grids. The MAE is computed against *Oracle*.

formance only degrades mildly and the variance across seeds remains stable, demonstrating that Twin is largely insensitive to grid sparsity. This robustness is particularly relevant in practical scenarios where dense LR–WD sweeps are infeasible due to computational constraints.

**Early stopping.** In Table 5, we analyze the impact of the early stopping scheduler. As visible, Twin effectively accommodates $HB_{25\%}$ or $HB_{12\%}$. Practitioners could safely default to either of the two, with $HB_{25\%}$ slightly ahead. The drop in performance for RNX101 with $HB_{12\%}$ is primarily due to the aggressive early-stopping budget, which allows only four out of the 36 trials ($6 \times 6$ grid) to complete, thereby increasing the likelihood of prematurely terminating high-performing configurations.

| | Small Datasets | | | Medical Datasets |
|---|---|---|---|---|
| Scheduler | EN-B0 | RN50 | RNX101 | RN18 |
| FIFO | **69.5** | **76.2** | **75.6** | 83.1 |
| $HB_{25\%}$ | **69.5** (+0.0) | **76.2** (+0.0) | **75.6** (+0.0) | **83.4** (+0.3) |
| $HB_{12\%}$ | **69.5** (+0.0) | 75.1 (-1.1) | 70.4 (-5.2) | 83.3 (+0.2) |

Table 5: **Early stopping.** Twin works robustly with $HB_{25/12\%}$ as well as FIFO. We report the average balanced test accuracy (%) on small datasets and test accuracy (%) on medical datasets.

Recall that for Twin, the validation metric used by the scheduler is the train, not the validation loss. Therefore, a general remark for smaller grids (i.e., $\leq 6 \times 6$) is to balance the amount of early stopping with the overall number of trials, since in some cases the loss landscape may become too noisy, preventing Twin from optimally selecting HPs. Guided by our empirical results, we suggest a minimum of 25% ending trials when small grids and early stopping are employed.

**Optimizers and schedulers.** In all experiments throughout the paper, we employed SGD with momentum (SGDM) and LR cosine scheduler as standard practice in deep learning. In this paragraph, we elaborate on the possibility of using different optimization setups. In particular, we test plain SGD in six configurations involving ConvNets. We also test a piece-wise LR scheduler. Finally,

| Dataset | Model | Optim. Setup | Perf. | (M)AE |
|---|---|---|---|---|
| {C10, C100, ES, I2018, c10, CM} | {WRN-16-10, RN20, RN50} | SGD | 75.0 | 1.8 |
| | | SGDM | 75.6 | 1.2 |
| C100 | CVT-7/8 | AdamW | 67.7 | 1.1 |
| C10 | MLP-4-256 | Adam | 65.5 | 1.8 |
| C10 | RN20 | SGDM (piece-wise) | 91.6 | 1.1 |
| | | SGDM (cosine) | 91.8 | 0.9 |

Table 6: **Different optimization setups.** Twin works successfully beyond SGDM (SGD, Adam, AdamW) and cosine LR scheduler (piece-wise). Performance is the (balanced) test accuracy. The (M)AE is computed against the *Oracle* baseline.

we experiment with either Adam or AdamW, two equally popular optimizer choices. In Table 6, we observe that Twin also closely follows the *Oracle* in terms of (mean) absolute error (M)AE in such alternative optimization setups.

# 5 Related Work

**Hyper-parameter tuning.** There is a vast literature tackling the problem of HP tuning for deep networks (Yu & Zhu, 2020), including works on implicit differentiation (Lorraine et al., 2020), data augmentation (Cubuk et al., 2019), neural-architecture search (Elsken et al., 2019), general-purpose schedulers (Li et al., 2017; 2020). A line of work investigated the zero-shot transfer of HPs across different model dimensions (Yang et al., 2021; Bordelon et al., 2024), data scales (Yun et al., 2020; 2022), and batch sizes via the linear scaling rule (Goyal et al., 2017). However, only a few studies explore HPO without employing validation

sets, mainly focusing on learning invariances. Bayesian methods either fail to scale to relatively simple tasks (Schwöbel et al., 2022) or modest network sizes (Immer et al., 2022). Alternatively, previous attempts either make strong assumptions in advance (Benton et al., 2020) or introduce complexity through model partitioning and computational overheads (Mlodozeniec et al., 2023). Unlike such methods, Twin easily scales to increased model and data sizes and simplifies the LR-WD optimization pipeline.

**Predicting generalization** Most of the work on predicting generalization in deep learning has focused on assessing the generalization gap, via several complexity metrics based on model parameters, the training set, and distributional robustness, among others (Keskar et al., 2016; Chuang et al., 2021; Smith & Le, 2017; Dziugaite & Roy, 2017; Dinh et al., 2017; Dziugaite et al., 2020; Jiang et al., 2020; Corneanu et al., 2020; Jiang et al., 2018; Neyshabur et al., 2017). For differences with Twin, refer to Section 2. Another popular line of research has long studied the sharpness of minima as a proxy for generalization in deep learning (Hochreiter & Schmidhuber, 1994; Jiang et al., 2020; Andriushchenko et al., 2023). The intuition from these studies underpinned methods like SAM (Foret et al., 2021). Unlike sharpness, we propose a separability-informed decision rule that uses the training loss and parameters' norm to predict optimal LR and WD configurations.

**Early stopping without validation sets.** Several works aim to design stopping criteria that remove the need for a held-out validation set while preventing overfitting. Approaches exploit training dynamics, such as gradient signals (Forouzesh & Thiran, 2021), the convergence behavior of parallel models (Vardasbi et al., 2022), or neuron's transfer functions (Dalmasso et al., 2025). More recent methods address robustness to label noise (Yuan et al., 2025a), or introduce instance-level criteria to reduce unnecessary updates (Yuan et al., 2025b). Bayesian perspectives have also been explored, framing early stopping as posterior sampling along the optimization trajectory (Mahsereci et al., 2017). For differences with Twin, refer to Section 2.

## 6 Conclusions

We introduced Twin, a simple yet effective HP tuning approach that reliably predicts LR and WD of deep homogeneous classifiers without using validation sets across dataset sizes, imaging domains, architectures, model scales, and training setups. Beyond the practical algorithm, which simplifies model selection, our paper introduced an analysis connecting margin maximization in homogeneous networks with an empirical train-test loss scaling law, yielding a regime-dependent understanding of when training loss or parameters' norms reliably predict test performance. We hope that this work will stimulate further investigation into validation-free approaches for predicting generalization in deep networks, informed by training dynamics and train-objective scaling behavior.

**Limitations and future work.** Like any other new approach, Twin comes with its own limitations. For instance, Twin currently optimizes LR and WD, which are essential yet represent a restricted subset of HPs. We expand upon them in Appendix F. Interesting future directions (Appendix G) include grounding theoretical understanding, refining generalization indicators, extending the pipeline to support additional samplers and schedulers, testing Twin on other tasks and modalities, and extending it to other regularization strategies that explicitly affect train dynamics (e.g., data augmentation shown by Balestriero et al., 2022).

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

## A  Estimation of Validation Error Variance

To estimate optimal HPs such as $\alpha$ and $\lambda$, we typically rely on a validation set as a surrogate for test performance. Standard approaches either split $\mathcal{D}_{\text{train}}$ into $\widehat{\mathcal{D}}_{\text{train}}$ and $\mathcal{D}_{\text{val}}$, with $|\widehat{\mathcal{D}}_{\text{train}}| = n - m$ and $|\mathcal{D}_{\text{val}}| = m$ or collect an external $\mathcal{D}_{\text{val}}$. Let $\delta = \mathbb{E}[\ell(f_\theta(x), y)]$ and $\sigma^2 = \text{Var}[\ell(f_\theta(x), y)]$ denote the expected and variance of generalization error over unseen samples. Let the validation loss be $\bar{\mathcal{L}}(f_\theta, \mathcal{D}_{\text{val}}) = \frac{1}{m} \sum_{i=1}^{m} \ell(f_\theta(x_i), y_i)$. By applying the linearity of expectation and the defined relationships, we derive $\mathbb{E}[\bar{\mathcal{L}}(f_\theta, \mathcal{D}_{\text{val}})] = \delta$, $\text{Var}[\bar{\mathcal{L}}(f_\theta, \mathcal{D}_{\text{val}})] = \frac{\sigma^2}{m}$, and $\text{Std} = \mathcal{O}(1/\sqrt{m})$. This implies that the validation estimate deviates from the test error by a term depending on $m$: $\mathbb{E}[\bar{\mathcal{L}}(f_\theta, \mathcal{D}_{\text{val}})] = \delta \pm \mathcal{O}(1/\sqrt{m})$. Hence, a small $m$ leads to a noisy estimate of generalization performance for HP selection, presenting a major drawback for practitioners.

## B  Empirical Validation of Train-Test Loss Scaling Law

In the main text, we demonstrated that our proposed parametric model (Equation (9)) accurately captures the phase transitions and train-test loss dynamics for both a controlled theoretical setup from Vysogorets et al. (2024) (LeNet300-100 on FashionMNIST) and one of our HPO search problems (ResNet50 on CUB). To provide a broader empirical validation of our power-law formulation beyond this targeted comparison, we present the fitted scaling laws and quantitative goodness-of-fit metrics across all 37 dataset-model configurations evaluated in our study.

**Qualitative goodness-of-fit.**  In Figure 10, we show the individual bias-variance scaling laws fitted on $\bar{\mathcal{L}}_{\text{HPO}}(\mathcal{D}_{\text{train}})$ and $\bar{\mathcal{L}}_{\text{HPO}}(\mathcal{D}_{\text{test}})$. The empirical scaling law smoothly adapts to highly diverse learning behaviors. Depending on the specific problem setup in terms of dataset, model, data augmentation, etc., the HPO configurations span different segments of the theoretical curve. Some setups capture the full characteristic U-shape (bridging both the divergence and lazy learning regimes), while others remain predominantly in one of the two. This experimental evaluation spans a vast array of dataset domains (e.g., from natural to medical images) and data types (e.g., RGB and multi-channel images), a wide range of dataset sizes (from tens to thousands of images per class) and model sizes (from a few thousand to $\sim$90M trainable parameters), and varying data augmentation strengths (from simple flipping to RandAugment).

**Quantitative goodness-of-fit.**  To move beyond qualitative visual inspection, we compute the global goodness-of-fit for our proposed parametric formulation. Figure 7 reports the quantitative results of this analysis across all evaluated configurations. On the left, the aggregated scatter plot of the predicted test losses versus the actual test losses demonstrates a strong alignment along the perfect fit diagonal ($y = x$), scoring a global $R^2$ of 0.896 and successfully holding over multiple orders of magnitude. On the right, we report the distribution of the log-space $R^2$ scores across the 37 configurations. The distribution is heavily skewed toward one (mean $R^2 = 0.792$, median $R^2 = 0.955$), indicating that the fit is tight for the vast majority of setups. We emphasize that while these results do not establish absolute mathematical universality for all conceivable neural network optimizations, they provide compelling evidence that the observed train-test relationship is not an artifact of a specific task or architecture. Instead, the relation defined in Equation (9) is remarkably consistent across a wide range of practical deep learning settings tested in this work.

## C  Modeling Approximations

In our derivation of Section 3.2, we made some simplifying assumptions concerning the noise in the scaling law, the smoothed margin, and exact $k$-homogeneity, which we discuss further here.

**1. Noiseless scaling law.** Although the mathematical modeling relied on the noiseless scaling law for clarity, a more realistic empirical relation between training and test loss is better described by the heteroskedastic form $\mathcal{L}_{\text{test}} = \xi(\mathcal{L}_{\text{train}}) + \epsilon$, where $\epsilon$ is a zero-mean noise term whose scale often depends on $\mathcal{L}_{\text{train}}$ (e.g., $\text{Std}[\epsilon] = \epsilon_0 \, \mathcal{L}_{\text{train}}^{-e}$ with $e \geq 0$). Empirically, this heteroskedastic structure leads to increased variability in $\mathcal{L}_{\text{test}}$ as models approach overfitting. Such fluctuations may slightly blur the exact transition between the different optimization regimes and cause local irregularities in the observed ($\mathcal{L}_{\text{train}}, \mathcal{L}_{\text{test}}$) relation. We hypothesize that tailored variance reduction techniques may further improve validation-free prediction.

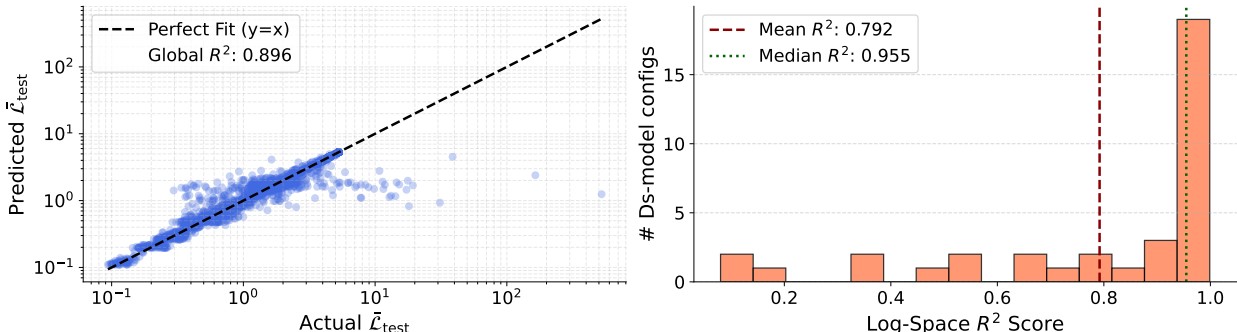

Figure 7: **Quantitative fit of train-test loss scaling laws over 37 dataset-model configurations.** Global aggregated fit between actual test losses and predicted test losses by the scaling law show a strong alignment over multiple orders of magnitude (**left**). The fit is tight for most of the tested dataset-model configurations being the distribution skewed towards one (**right**).

**2. Smoothed normalized margin.** Our analysis relies on the smoothed normalized margin $\tilde{\gamma}$, which introduces a quantization gap from the normalized margin $\bar{\gamma}$. The bound $\tilde{\gamma}(\theta) \leq \bar{\gamma}(\theta) \leq \tilde{\gamma}(\theta) + \frac{\log N}{\rho^k}$ implies that, e.g., when the normalized margin $\bar{\gamma}$ is positive, the smoothed surrogate $\tilde{\gamma}$ can be still negative, adding a small, norm- and dataset-dependent noise to the separability transition, particularly for low-norm networks (small $\rho$) trained on big datasets (large $N$). To address the presence of such noise, we slightly adapted the separability threshold $\tau_{\text{acc}}$ and showed the effectiveness of this tweak in Section 4.3.

**3. Homogeneity of degree $k$.** Commonly-used neural models are usually only approximately homogeneous of degree $k$ since different architectural components within a model, such as the final linear layer versus an attention block, create deviations from exact $k$-homogeneity, making models multi-homogeneous (Li et al., 2022). Lyu & Li (2019), extend the formula expressed in Equation (8) to the multi-homogeneous case and show that the relation holds by substituting $\rho^k$ with $\prod_i \rho_i^{k_i}$, where $\rho_i$ represents the norm of each multi-homogeneous part. Our simplification, therefore, introduces a typically minor additional source of noise in the mapping between norm dynamics, margin evolution, and train-loss decay, with the possibility of extending the analysis to the multi-homogeneous setting left for future work.

# D   Implementation Details

## D.1   Small Datasets

We train all networks with batch sizes of 10 samples, given the better generalization performance of small batch sizes in small-sample regimes (Brigato et al., 2021; 2022). The training iterations are drawn from (Brigato et al., 2022), with a minimum of 25,000 for the smaller datasets and a maximum of roughly 120,000 for CUB. We employ standard image pre-processing transformations, including data augmentation utilizing random crops and horizontal flipping, also following Brigato et al. (2022). More precisely, all input images were normalized by subtracting the channel-wise mean and dividing by the standard deviation computed on the training splits. For datasets with a small, fixed image resolution, i.e., ciFAIR-10 and EuroSAT, we perform random shifting by 12.5% of the image size and horizontal flipping in 50% of the cases. For all other datasets, we apply scale augmentation using the `RandomResizedCrop` transform from PyTorch[1] as follows: A crop with a random aspect ratio drawn from $[\frac{3}{4}, \frac{4}{3}]$ and a dataset-dependent area between $A_{\min}$ and 100% of the original image area is extracted from the image and then resized to $224 \times 224$ pixels. We use $A_{\min} = 20\%$ for CLaMM and $A_{\min} = 40\%$ for CUB and ISIC 2018. For ISIC 2018 and EuroSAT, we perform random vertical flipping in addition to horizontal flipping since these datasets are completely rotation-invariant, and vertical reflection augments the training sets without drifting them away from the test distributions. On

---

[1]https://pytorch.org/vision/stable/transforms.html#torchvision.transforms.RandomResizedCrop

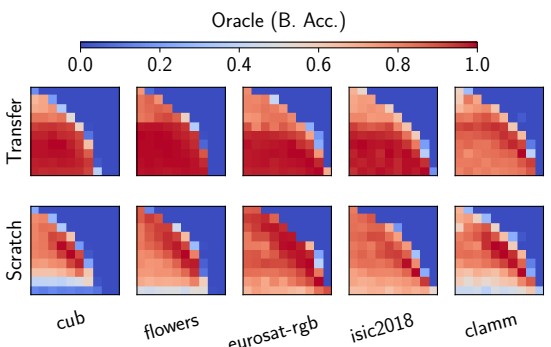 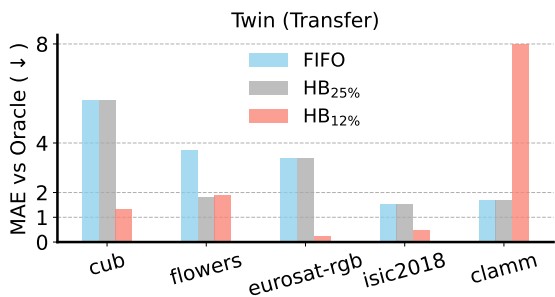

Figure 8: **(Left)** Normalized balanced accuracy on the LR-WD space of the *Oracle* with ImageNet pre-trained (top) or from-scratch RN50 (bottom). Optimal generalization occurs at lower regularization (bottom left area) when pre-training/downstream features overlap (i.e., CUB, Flowers, EuroSAT-RGB, ISIC 2018), a trend not observed in CLaMM (hand-written documents). **(Right)** Twin plus early stopping (EN-B0, RN50, and RNX101) counteracts this LR-WD landscape change by scoring a low MAE. **Domain overlap dictates the most suitable scheduler choice**: aggressive early stopping (HB$_{12\%}$) is preferred with high class/feature overlap and more moderate or absent early stopping in the opposite case (HB$_{25\%}$ or FIFO).

CLaMM, in contrast, we do not perform any flipping since handwritten scripts are not invariant, even against horizontal flipping.

## D.2 Medical Images

We fixed the batch size to 50 samples and aimed for roughly 50,000 training steps, accordingly adapting the number of epochs per dataset. For data augmentation, we perform random translations of a maximum of four pixels per side as done by He et al. (2016). We keep the same Twin configurations tested in the small-dataset experiments.

## D.3 Natural Images

We split the original training sets into 80%-20% to perform the HP selection for the SelVS baseline. All convolutional and feed-forward networks are optimized with a batch size of 50 for 100 epochs. As discussed in the paper, we vary the data augmentation strength from base to medium to strong and represent it with $\{+, ++, +++\}$. The lowest level $(+)$ corresponds to the plain 4-pixel translation plus random horizontal flipping as initially proposed by He et al. (2016). The second level uses the more aggressive RandAugment (RA) strategy (Cubuk et al., 2020) with default parameters (N = 2, M = 9), except for WRN-40-2 (N = 3, M = 4) following Cubuk et al. (2020). Finally, the highest level of data augmentation $(+++)$ implemented even stronger RA parameters (N = 2, M = 14), MixUp (Zhang et al., 2017) ($\alpha_{mixup} = 0.8$) and CutMix (Yun et al., 2019) ($\alpha_{cutmix} = 1.0$). The augmentation level $+++$ is typical for training ViTs, which are more difficult to train given the lack of inductive biases. Following the original work of Hassani et al. (2021), we also add to the CCT and CVT training pipeline label smoothing of 0.1 and a cosine annealing scheduler with a learning rate warm start. The warm start ends at 10% of the total number of iterations. To train ViTs, we employed batch sizes of 128 images for 500 epochs and set the LR and WD intervals to $[10^{-6}, 10^{-1}]$.

## E    Transfer Learning Experiments

### E.1    Domain Overlap Affects LR-WD Landscapes and Train-Test Loss Scaling Law

When dealing with small datasets, it is common practice to start from a network pre-trained on a larger amount of data (e.g., ImageNet Russakovsky et al., 2015). Therefore, we also experiment with transfer

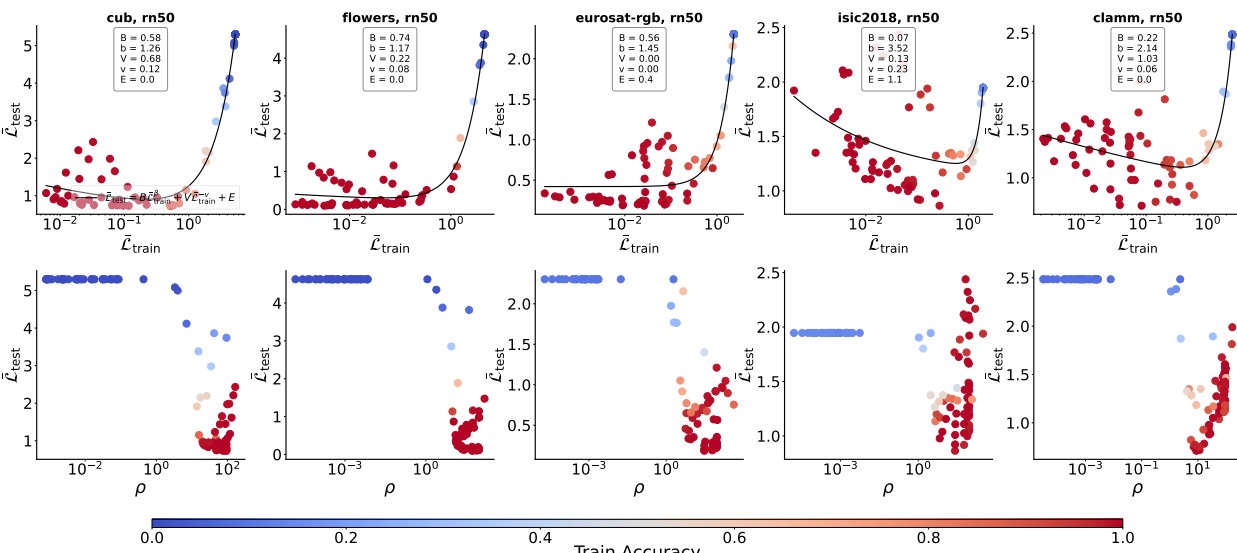

Figure 9: **Train-test loss scaling laws and norm dynamics in transfer learning.** Five experiments with RN50 illustrating how the fitted scaling law (**top**) and test loss vs. norm relation (**bottom**) evolve across learning regimes when performing transfer learning. Each point corresponds to an LR–WD configuration, color-coded by training accuracy. The overlap between pre-training/downstream classes is significant for CUB and Flowers. There is also some high-level feature transfer for EuroSAT RGB and ISIC 2018 datasets. In these cases, the train-test scaling law is noisier (top row) and hence reduces monotonicity among the test loss and parameters' norm (bottom row). However, this is not the case for CLaMM (low feature overlap), where the scaling law resembles the training-from-scratch cases shown in Figure 3.

learning and repeat a subset of the optimization runs with ImageNet checkpoints. In Figure 8 (left), we notice that the generalization of networks as a function of the LR-WD space may differ from when training from scratch, and the main cause regards the overlapping between the source and target domains. Expectedly, with a strong class (CUB and Flowers) or feature (EuroSAT RGB, ISIC 2018) overlap, the best generalization region shifts towards smaller regularization. In these cases, the overlap between pre-trained and downstream representations induces a noisier relation between training and test loss, which weakens the monotonic correspondence between test loss and parameters' norm (Figure 9, top left). As a consequence, the fitted scaling law becomes less reliable and Twin underperforms, as reflected by the higher MAE (up to ∼5%) observed for CUB, EuroSAT RGB, and ISIC 2018 (Figure 8, right).

To mitigate this effect, we employ early stopping (HB$_{12\%}$) to prune heavily regularized configurations whose training loss decays slowly. As shown in Figure 8 (right), Twin combined with HB$_{12\%}$ substantially reduces the MAE relative to the *Oracle*, reaching ≤ 1% for CUB, EuroSAT RGB, and ISIC 2018. In contrast, for CLaMM (hand-written documents), which exhibits neither class overlap nor strong feature alignment with the pre-training data, the LR-WD landscape remains largely unchanged by transfer learning (Figure 8, left). In this setting, the train-test scaling law remains well-structured (Figure 9), enabling Twin to identify competitive HP configurations with both the FIFO and HB$_{25\%}$ schedulers, achieving an MAE below 2%.

In summary, when applying transfer learning, it is critical to consider the level of domain overlap to select the most suitable scheduler for the Twin configuration. Extending Twin to explicitly model the interaction between transfer learning and scaling-law noise remains an important direction for future work.

## F  Limitations

First, although Twin is grounded in a mathematical analysis that combines theoretical results on margin-maximization dynamics in homogeneous classifiers (Lyu & Li, 2019) with an empirical train-test loss scaling

law, the resulting methodology necessarily relies on a set of simplifying assumptions. In particular, the analytical derivations abstract away sources of stochasticity and architectural heterogeneity that may affect real training dynamics. As a consequence, there may exist regimes where the assumed monotonic relations between training loss, parameters' norm, and generalization are weakened or partially violated. One such example arises in transfer learning, where optimization dynamics may enter qualitatively different regimes depending on the degree of class or feature overlap between pre-training and downstream tasks (Appendix E). As we empirically demonstrated, strong overlap can introduce additional noise in the train–test loss scaling relation, which we mitigate through more aggressive early stopping. Nevertheless, a more comprehensive theoretical characterization of these regimes remains to be explored. Additionally, our formal derivations are grounded in the margin-maximization dynamics of exponential-tailed losses (e.g., cross-entropy) (Lyu & Li, 2019). While this covers the most prevalent classification setups, the theory does not directly extend to regression tasks or alternative loss functions (such as $L_2$), where the lack of margin-driven norm growth may necessitate different analytical tools.

From the perspective of empirical scope, Twin currently focuses on tuning LR and WD. While these HPs are both fundamental and among the most commonly tuned by practitioners, they represent only a subset of the broader HP landscape. Moreover, our experimental evaluation is restricted to image classification tasks. Given that margin maximization and train-test loss scaling laws are domain-agnostic, our results on vision benchmarks provide a promising foundation for future extensions to other modalities.

Finally, regarding HPO design choices, Twin currently operates over a grid-based LR–WD search combined with simple scheduling strategies. While grid search remains a widely adopted baseline in practice (Kornblith et al., 2019; Steiner et al., 2022), it can be inefficient compared to more adaptive sampling strategies. We partially address this through early-stopping schedulers ($\text{HB}_{X\%}$) and by demonstrating robustness to grid sparsity (Section 4.3). Nonetheless, integrating Twin with more flexible or adaptive search mechanisms is a natural direction for improvement.

These limitations open up several promising future directions, which we outline next.

# G  Future Work

## G.1  Refinement of Generalization Indicators and Theoretical Understanding

While our analysis relies on a deterministic train-test loss scaling law for clarity, empirical train–test relations are inherently noisy, especially in overfitting regimes. A promising direction is therefore the development of tailored variance-reduction techniques that stabilize validation-free generalization signals. Such techniques may be architectural, for instance, residual connections are known to induce approximately linear variance growth with depth (Brock et al., 2021). Alternatively, they may arise from explicit loss penalties that regularize train loss fluctuations. Understanding how architectural choices and regularization schemes modulate the increasing term in the scaling law could substantially improve the reliability of validation-free predictors.

Characterizing the role of the smoothed normalized margin used in our analysis would also be interesting. Quantifying how the gap between the smoothed surrogate and the true normalized margin affects the separability transition, especially as a function of dataset size and parameters' norm, would sharpen the theoretical guarantees of the method. Closely related is the extension of the analysis to multi-homogeneous networks, where different components contribute distinct degrees of homogeneity. Furthermore, designing architectures that are explicitly $k$-homogeneous (Li et al., 2022) may offer a principled way to strengthen the connection between margin maximization, norm growth, and generalization.

On the other hand, transfer learning remains a particularly compelling setting in this context. Pre-training and fine-tuning can induce nontrivial interactions between weight decay, parameters' norms, and effective margins, often resulting in noisier or distorted bias–variance scaling laws. Developing a theoretical and empirical understanding of how domain or feature overlap influences these dynamics could extend validation-free prediction to a broader class of practical scenarios, including large-scale pre-trained models.

Finally, Twin currently utilizes margin-maximization to link training observables to generalization in classification. While this theoretical grounding is specific to exponential-type losses (Lyu & Li, 2019), the empirical

success of Twin suggests that the underlying train-test loss scaling laws may be more universal due to its link to optimization regimes. A compelling direction for future research is to investigate whether analogous validation-free indicators can be identified for regression tasks or alternative loss functions where the optimization dynamics may differ.

### G.2 Extension of Empirical Scope and Practical Tuning Pipeline

We see the broadening of the empirical scope as another important avenue. Testing Twin on different modalities and beyond image classification will strengthen its generality.

Moreover, the current formulation of Twin focuses on LR and WD. Extending it to other regularization strategies that directly affect train dynamics could make the method more broadly useful to practitioners. For instance, Balestriero et al. (2022) has shown that data augmentation has a direct impact on the magnitude of the parameters' norm. This suggests the possibility of tuning the data augmentation strength by following the same principle applied for WD. More generally, adapting Twin to optimize multiple HPs simultaneously, while maintaining simplicity and effectiveness, is a valuable research direction.

Finally, a valuable direction is the practical improvement of the pipeline in terms of efficiency and supported components. Replacing the current grid search and schedulers with more advanced strategies could further reduce computational costs while maintaining or even improving performance. This line would require the development of robust filtering modules to identify the fitting regions, due to more noisy loss landscapes in the HP search space. We demonstrated in our sensitivity analysis on grid density that this is a viable direction, given that Twin, in its current form, supports grid densities of varying sparsity.

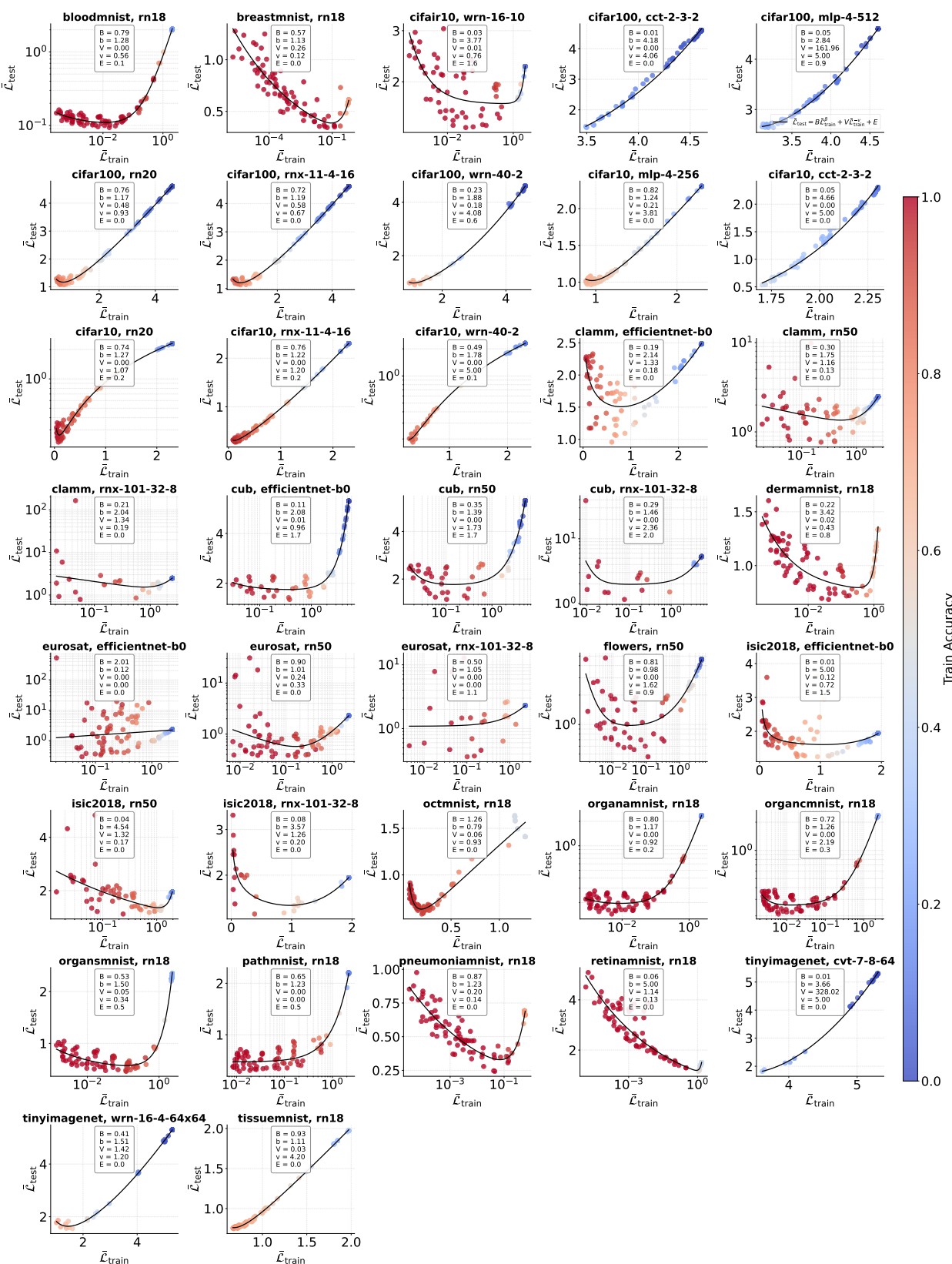

Figure 10: **Qualitative view of train-test scaling laws over 37 dataset-model configurations.**

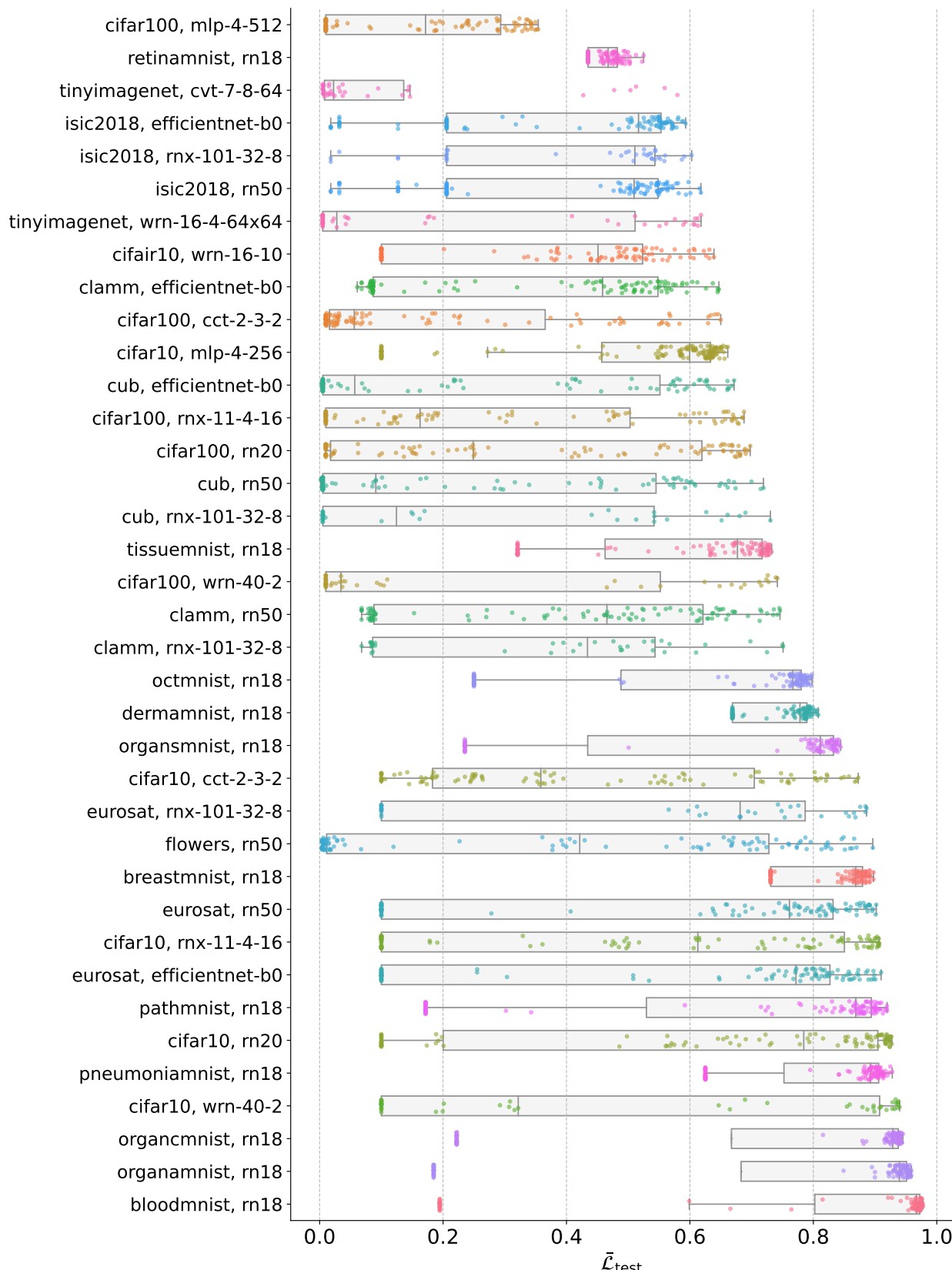

Figure 11: **Test accuracy distributions over 37 dataset-model configurations.**

