# OpenReview forum: "Twin: Tuning Learning Rate and Weight Decay of Deep Homogeneous Classifiers without Validation"
_TMLR — Accepted by TMLR_

### Review · Reviewer_ZyEs · 2026-03-08

**Summary Of Contributions:**

**Summary**

The paper proposes a hyperparameter optimization method, Twin, that uses only the training set. The authors focus on tuning the learning rate and weight decay for classification tasks by leveraging the bias-variance trade-off. They demonstrate that their hyperparameter selection achieves results comparable to tuning on a validation dataset.


**Strength**

* The paper is generally well-written and easy to understand.
* Hyperparameter optimization without a validation set is an important topic, especially for benchmarks lacking a validation set or when data is too limited to construct one.
* The proposed method, Twin, achieves performance comparable to an oracle (which accesses the test set) across various datasets (Figure 3).


**Weakness**

* The proposed method is limited to classification tasks using cross-entropy loss. It would strengthen the paper to demonstrate that the derivation in Section 3 is loss- and task-agnostic. For example, the authors could evaluate a regression task using L2 loss or a classification task with a different loss function (or auxiliary loss).
* Equation (9) is the most important equation in the paper, connecting test loss and training loss. However, it lacks a theoretical proof. It appears to be empirically defined and may not fit well across all tasks.
* The notation in Lines 12-16 of Algorithm 1 is flawed. The authors define the mask $\mu$, multiply it by $\Theta$, and apply an argmin. Since $\Theta \geq 0$, the argmin will always return 0.
* Several descriptions are vague. For example, regarding the statement `To avoid such degenerate solutions, we filter out any configuration achieving less than 15% training accuracy,` the authors should explain how the 15% threshold was chosen. The term `trivial accuracy` on page 9 also requires clarification.
* Although the results look promising, the authors need to show the accuracy distribution across all hyperparameter configurations to demonstrate the actual difficulty of the tuning task. Otherwise, readers might assume the strong results are simply due to the models being robust to hyperparameter changes. This omission affects the interpretation of Tables 1-5 and Figure 3.
* Section 4 should be restructured for better flow. Currently, it presents results, introduces the experimental setting, and then returns to the results.
* The authors should clarify what makes the observation "interesting" when using the term `interestingly` (page 13).
* In the sentence `Even with 60%–80% of the grid cells removed, performance only degrades mildly and exhibits low variance across seeds,` the term `low variance` is subjective. In Figure 5, the variance does not appear to change significantly between 40% and 80%.

**Questions**

* Why is the absence of scheduling referred to as FIFO (presumably First-In-First-Out) on page 8?
* Why did the authors conduct 100 trials for RN50 and EN-B0, but only 36 trials for RNX101? It would be better to unify the number of trials.
* What are the exact criteria for early stopping? In Appendix D, the authors state, `we employ early stopping to prune heavily regularized configurations whose training loss decays slowly,` but a precise mathematical or threshold definition is needed.
* How is the strength of data augmentation measured on page 12?

**Audience:**

Yes

**Audience Explanation:**

Hyperparameter optimization without a validation set is an important topic, especially for benchmarks lacking a validation set or when data is too limited to construct one.

**Broader Impact Concerns:**

I do not have any concerns regarding ethical implications.

**Claims And Evidence:**

No

**Claims Explanation:**

As mentioned above, the authors need to clarify the equations and results.

**Requested Changes:**

* In Page 6, parameter => parameters
* Citep is used instead of citet in several places such as (e.g., (Forouzesh & Thiran, 2021)), (Jiang et al., 2018) in page 3., (Brigato et al., 2022) in Section 4.2.1,

---

> ### Author Response · Authors · 2026-04-13
>
> We thank the reviewer for the time invested in reviewing our paper and appreciate the several inputs provided during the review, which have improved our manuscript, as explained below. We address the requested changes and asked questions as follows.
>
> > The proposed method is limited to classification tasks using cross-entropy loss. It would strengthen the paper to demonstrate that the derivation in Section 3 is loss- and task-agnostic. For example, the authors could evaluate a regression task using L2 loss or a classification task with a different loss function (or auxiliary loss).
>
> We thank the reviewer for this suggestion and agree that extending Twin to be entirely loss- and task-agnostic is an exciting prospect. However, the theoretical foundation of our approach (Section 3) is, to the best of our knowledge, tied to the properties of exponential-tailed loss functions. Indeed, as established by Lyu and Li (2019), the monotonic increase in the smoothed normalized margin is driven by the asymptotic behavior of exponential-type losses (such as cross-entropy and logistic loss). Adapting this specific derivation to an $L_2$ loss for regression is mathematically non-trivial, since $L_2$ has different norm dynamics and implicit regularization behavior. Because of this reliance on exponential-tailed losses, we deliberately scoped our paper and title specifically to "Classifiers." Furthermore, as noted by Lyu and Li (2019), this assumption is not restrictive within this scope, as it encompasses the most popular classification losses used in deep learning, with cross-entropy being the go-to loss for any deep classifier.
>
> That being said, we agree with the reviewer's intuition that the broader concepts may generalize. We hypothesize that the empirical train-test scaling laws we observe across learning rate and weight decay sweeps might hold, or have direct analogs in other optimization setups (e.g., regression with $L_2$ loss), even if the margin-maximization mechanism differs. However, this extension would require significant additional mathematical modeling and empirical testing. To address the reviewer's point and better contextualize our contributions, we have expanded the Limitations/Future Work sections (Appendix F and G of the revised manuscript) to more broadly discuss this limitation and to highlight the exploration of Twin beyond exponential-like functions and classification tasks as an important direction for future work.

---

> ### Author Response · Authors · 2026-04-13
>
> > Equation (9) is the most important equation in the paper, connecting test loss and training loss. However, it lacks a theoretical proof. It appears to be empirically defined and may not fit well across all tasks.
>
> We completely agree that this formulation requires stronger justification. Prompted by the reviewer and reviewer **Bn2y**'s feedback, we have expanded both the theoretical discussion and empirical evidence. In particular, we emphasize that Equation (9) should be interpreted as an empirical scaling law rather than a universal theorem, and we show that it is consistent with optimization regimes recently formalized in deep learning theory.
>
> **Optimization regimes and the Goldilocks zone.** We believe that Equation (9) models the learning process across three distinct optimization regimes governed by the local geometry of the loss. Recent theoretical research concerning the "Goldilocks zone" provides the mechanistic explanation for these regimes. Fort & Scherlis (2019) discovered that a large excess of positive curvature and local convexity of the loss Hessian is associated with highly trainable initial points (i.e., parameter norms) located in this Goldilocks zone. More recently, Vysogorets et al. (2024) rigorously derived that this excess of positive curvature disappears at hyperparameter extremes. It is removed by saturated softmax on one end (resulting in a lazy learning regime) and by vanishing logit gradients on the other (resulting in divergence). While the Goldilocks zone was originally linked to initialization scale by Fort & Scherlis (2019), Vysogorets et al. show it is equally achievable via softmax temperature tuning, which has an equivalent effect on loss curvature in homogeneous networks. In our HPO setting, varying learning rate and weight decay appear to traverse analogous optimization conditions, thereby inducing comparable transitions between regimes. From this perspective, Eq. (9) should be viewed as a phenomenological model summarizing how train and test losses behave across these regimes
>
> **Comparison with Vysogorets et al. optimization regimes.** To empirically support this connection, we replicated the exact setup used by Vysogorets et al. in their Section 5 experiment, using their original code. Precisely, we trained a homogeneous feedforward LeNet300-100 with full gradient descent on FashionMNIST for 20,000 iterations, sweeping learning rate and initialization scale. We compared this setup directly to one of our own dataset-model configurations, i.e., ResNet50 on CUB, sweeping learning rate and weight decay. As shown in Figure 2 of the revised manuscript:
>
> - Both setups exhibit very similar staircase-like heatmaps for training and testing accuracy, indicating sharp transitions in trainability consistent with the regime structure described by Vysogorets et al.
> - When plotting train vs test losses the replicated setup yields a qualitatively similar U-shaped relationship, closely matching the functional form of Eq. (9).
>
> These results suggest that the scaling law we observe is not an artifact of a specific architecture or dataset, but rather reflects how optimization dynamics organize homogeneous classifiers trained with cross-entropy loss across training regimes.
>
> **Broader empirical validation of the power law form.** Beyond this controlled comparison, we provide extensive empirical validation across our 37 dataset–model configurations. In the revised manuscript, Figure 10 (Appendix B) shows qualitative fits of the scaling law, and Figure 7 (Appendix B) reports quantitative goodness-of-fit, with a global $R^{2}$ of 0.896.
> This experimental evaluation spans a vast array of dataset domains (e.g., from natural to medical images) and data types (e.g., RGB and multi-channel images), a wide range of dataset (from tens to thousands of images per class) and model sizes (from few thousands to ~90M trainable parameters), and varying data augmentation strengths (from simple flipping to RandAugment). We emphasize that this does not establish universality, but demonstrates that the observed train–test relation is remarkably consistent across a wide range of practical deep learning settings.
>
> **Power law forms in recent literature.** Finally, our choice of a power-law parameterization is supported by recent work on loss-to-loss scaling (Brandfonbrener et al. 2025; Mayilvahanan et al. 2025). These empirical scaling-law studies confirm that test loss can be expressed as a power-law function of train loss.
>
> Fort, Stanislav, and Adam Scherlis. "The goldilocks zone: Towards better understanding of neural network loss landscapes.", AAAI 2019
>
> Vysogorets, Artem, et al. "Deconstructing the goldilocks zone of neural network initialization.", ICML 2024
>
> Brandfonbrener, David, et al. "Loss-to-loss prediction: Scaling laws for all datasets”, TMLR 2025
>
> Mayilvahanan, Prasanna, et al. "Llms on the line: Data determines loss-to-loss scaling laws”, ICML 2025

---

> > ### Author Response · Authors · 2026-04-13
> >
> > > The notation in Lines 12-16 of Algorithm 1 is flawed. The authors define the mask , multiply it by , and apply an argmin. Since , the argmin will always return 0.
> >
> > We thank the reviewer for catching this notational oversight. We have updated Algorithm 1 and the text to resolve this. Instead of a Hadamard product, we now use standard Boolean indexing notation to accurately reflect that the minimum is taken exclusively over the subset of masked configurations.
> >
> > > Several descriptions are vague. For example, regarding the statement To avoid such degenerate solutions, we filter out any configuration achieving less than 15% training accuracy, the authors should explain how the 15% threshold was chosen. The term trivial accuracy on page 9 also requires clarification.
> >
> > We thank the reviewer for pointing out these ambiguities. We have clarified both points in the revised manuscript.
> >
> > 1. 15% threshold: We did not actively tune this threshold. It was chosen a priori as a conservative, heuristic floor to exclude models that suffered from training collapse that failed to learn any meaningful signal, without penalizing models that merely underperformed.
> > 2. Trivial accuracy: We have updated the text to explicitly define "trivial accuracy" as performance equivalent to or slightly above random chance prediction.
> >
> > > Although the results look promising, the authors need to show the accuracy distribution across all hyperparameter configurations to demonstrate the actual difficulty of the tuning task. Otherwise, readers might assume the strong results are simply due to the models being robust to hyperparameter changes. This omission affects the interpretation of Tables 1-5 and Figure 3.
> >
> > We thank the reviewer for this constructive suggestion. We agree that establishing the baseline difficulty of the hyperparameter tuning task further contextualizes the significance of our results. To address this, we have generated a new figure (included as Figure 11 in the revised Appendix) that visualizes the test accuracy distribution across all evaluated hyperparameter configurations for the 37 dataset-model pairs.
> > As described in the main text (Section 4.2) and shown in the figure, the models are highly sensitive to changes in hyperparameters. Key observations from the new distribution plot include massive accuracy ranges (up to roughly 90 percentage points) and long-tailed distributions. Consequently, Twin's good performance is not an artifact of simple tuning tasks but rather a direct result of the method's ability to reliably predict good hyperparameter configurations.
> >
> > > Section 4 should be restructured for better flow. Currently, it presents results, introduces the experimental setting, and then returns to the results.
> >
> > We thank the reviewer for this feedback. Our intention with the current structure was to present the experiments in three distinct narrative phases. First, Section 4.1 serves as an isolated, motivating comparison against sharpness-based selection to demonstrate why a new validation-free approach is necessary. Once that premise is established, Section 4.2 broadens the scope to evaluate Twin against traditional validation-based pipelines across various domains (small datasets, medical imaging, and natural images). Because each domain requires distinct architectures and datasets, we kept the specific implementation details and their respective results together to prevent the reader from having to flip back and forth between a monolithic setup section and the results. Finally, Section 4.3 presents sensitivity analyses.
> > However, we understand how this repeated "setup-then-result" loop could feel disjointed without a proper preamble. To resolve this, we have added a clear roadmap at the beginning of Section 4 that explicitly outlines the structure for the reader.
> >
> > > The authors should clarify what makes the observation "interesting" when using the term interestingly (page 13).
> >
> > “Interestingly” referred to the takeaway expressed in the next sentence, i.e., “This behavior is consistent with our theoretical analysis”. To avoid any source of confusion, we removed it since it is not strictly necessary.
> >
> > > In the sentence Even with 60%–80% of the grid cells removed, performance only degrades mildly and exhibits low variance across seeds, the term low variance is subjective. In Figure 5, the variance does not appear to change significantly between 40% and 80%.
> >
> > We agree with the reviewer that "low variance" is a subjective descriptor. Our intended point was that the variance remains stable and does not escalate as the grid becomes increasingly sparse. This suggests that Twin does not become more volatile or sensitive to grid sparsity. We updated the text accordingly.

---

> > > ### Author Response · Authors · 2026-04-13
> > >
> > > > Why is the absence of scheduling referred to as FIFO (presumably First-In-First-Out) on page 8?
> > >
> > > We thank the reviewer for the question. We use the term "FIFO" (First-In-First-Out) because it is the standard nomenclature established by modern hyperparameter optimization frameworks (such as [Ray Tune](https://docs.ray.io/en/latest/tune/api/schedulers.html#fifoscheduler-default-scheduler), which we used, and [AutoGluon](https://auto.gluon.ai/0.0.15/api/autogluon.scheduler.html#autogluon.scheduler.FIFOScheduler)). In these software ecosystems, the baseline scheduler, which evaluates configurations sequentially to completion without applying early stopping or dynamic resource allocation, is explicitly defined as the FIFO scheduler. To maintain consistency with the technical terminology, we have hence retained the term FIFO, while explicitly stating its practical meaning in this context ("no early stopping") in parentheses alongside its first use (Section 3.3).
> > >
> > > > Why did the authors conduct 100 trials for RN50 and EN-B0, but only 36 trials for RNX101? It would be better to unify the number of trials.
> > >
> > > We thank the reviewer for the question. While unifying the number of trials would provide a more symmetric experimental setup, our decision to use 36 trials for RNX101 and 100 for the others was driven by two factors. First, due to computational constraints, RNX101 is significantly more resource-intensive to train than both RN50 and EN-B0. Scaling its grid to 100 trials would have incurred a prohibitive computational cost. Second, this discrepancy serves as a methodological validation: by evaluating Twin across different grid budgets (100 vs. 36 trials), we demonstrate that Twin is robust to varying search space sizes. This confirms that the efficacy of our approach is not dependent on a specific number of trials. We have briefly added this rationale to the experimental setup section to clarify this choice for future readers.
> > >
> > > > What are the exact criteria for early stopping? In Appendix D, the authors state, we employ early stopping to prune heavily regularized configurations whose training loss decays slowly, but a precise mathematical or threshold definition is needed.
> > >
> > > We thank the reviewer for the requested clarification. Early stopping (i.e., a custom HyperBand that ends X% of trials; see Section 3.3) is applied to the train loss signal for Twin because it is a validation-free method; we cannot use validation metrics to halt unpromising trials. The criteria for early stopping follow the HyperBand algorithm, so we did not introduce any custom threshold. We added a clarification sentence in Section 3.3. In Appendix D, early stopping referred to the scheduler employed, i.e., HB with 12% of ending trials. We added a more explicit reference there as well.
> > >
> > > > How is the strength of data augmentation measured on page 12?
> > >
> > > We thank the reviewer for this question. The strength of data augmentation is categorized into three discrete regimes, base (+), medium (++), and strong (+++), based on the combination of techniques applied. Specifically, the base level corresponds to standard translation and horizontal flipping, the medium level introduces RandAugment, and the strong level combines heavy RandAugment with MixUp and CutMix. As originally referenced on page 12, the exact hyperparameter configurations for each of these levels were detailed in Appendix C.3 (Appendix D.3 of revised manuscript). We deliberately chose to relegate these specifics to the appendix to maintain the narrative flow of the main text and avoid overwhelming the reader with low-level implementation details, while still ensuring the experimental setup is fully documented and reproducible.
> > >
> > > > Requested changes
> > >
> > > We applied the requested changes to the revised manuscript.

---

### Review · Reviewer_Bn2y · 2026-03-12

**Summary Of Contributions:**

## High-level Summary
This paper provides a validation dataset-free hyperparameter selection method that utilizes the composition of training dataset fitness (separability) and learned parameter norm as a predictor of test loss. The authors ground their method with a theoretical statement that test loss can be expressed as a function of training loss and an irreducible term, where the margin-indicated separability determines which quantity to use to predict test loss.

## Strengths and Weaknesses
The proposed method somehow works across multiple experiment setup showing effectiveness while relaxing data and resource requirements for the hyperparameter selection. However, their proposed theoretical statement is not sufficiently justified -- raising questions about the shown effectiveness is just a specific artifact in the considered setup. Moreover, the applicability is limited to classification models; cannot cover LLM training scenarios. Besides, the method relies on meta-hyperparameter ($\tau_{\text{acc}}$ tuning that requires some gold information on test set / validation set, making the proposal not truly validation-free.

**Audience:**

Yes

**Audience Explanation:**

Their proposed framework is inspired by interesting theoretical statements -- logit margin-based separability to determine the predictor for test loss, and adaptively utilizse a simple quantity such as training loss and parameter norm, and shows promising prediction performance.

I guess some audiences who work on training dynamics and optimization may find this work interesting and insightful.

**Broader Impact Concerns:**

This kind of automatic validation-free HPO method can be utilized by adversarial users to optimize their attack methods robustly under relaxed data requirements. However, I think it is not that remarkable to add as a separate Broader Impact Statement.

**Claims And Evidence:**

No

**Claims Explanation:**

## About their theoretical model
The most crucial statement by the authors in this paper is equation (9), in my opinion, which formulates the test loss as a function of the train loss and irreducible term; other formulations are borrowed from the existing works or are trivial.

However, their proposed formulation is not justified theoretically or empirically at all; even though the claim "test loss can be just predicted from train loss up to a specific degree." is a very aggressive one that needs to be thoroughly justified. I think equation (9) needs a lot of assumptions in data structure, loss type, or so, to make it hold.

Besides, in Eq (9), the authors also define the bias as an exponential function of train loss and variance as also an exponential function of train loss, which means that bias and variance have an exponential relationship as well. This is also hard to hold without some assumptions, but the authors do not provide any justification or assumptions for this.

While the experimental results they provide show the promise of the practical implementation of their method in predicting test loss, the theoretical model, which is a motivation of that practical method, is not sufficiently verified and supported.


## About their practical implementation
Although the authors claim that their method is validation-free, it is actually not truly validation-free, since the method requires defining "threshold $\tau_{\text{acc}}$" which can not be solely defined from the training dataset without auxiliary information.

According to their theoretical model, it should be set as 100% to simulate true separability. However, they set it 99% by (provably) checking the test data performance, which is hard to say truly validation-free.

**Requested Changes:**

The authors should provide a strong justification for their theoretical formation of test loss in Eq (9). This is the most important point I would want to bring to revert my recommendation on this paper.

Second, I would expect that the authors expand their experiments beyond the moderate scale image classification setups to justify that their $\tau_{\text{acc}}$=0.99 decision is sufficiently robust so that it is truly validation-free method.

---

> ### Author Response · Authors · 2026-04-13
>
> We thank the reviewer for the time invested in reviewing our paper and appreciate the several inputs provided during the review, which have improved our manuscript, as explained below. We address the requested changes as follows.
> > About their theoretical model
>
> We completely agree that this formulation requires stronger justification. Prompted by the reviewer and reviewer **ZyEs**'s feedback, we have expanded both the theoretical discussion and empirical evidence. In particular, we emphasize that Equation (9) should be interpreted as an empirical scaling law rather than a universal theorem, and we show that it is consistent with optimization regimes recently formalized in deep learning theory.
>
> **Optimization regimes and the Goldilocks zone.** We believe that Equation (9) models the learning process across three distinct optimization regimes governed by the local geometry of the loss. Recent theoretical research concerning the "Goldilocks zone" provides the mechanistic explanation for these regimes. Fort & Scherlis (2019) discovered that a large excess of positive curvature and local convexity of the loss Hessian is associated with highly trainable initial points (i.e., parameter norms) located in this Goldilocks zone. More recently, Vysogorets et al. (2024) rigorously derived that this excess of positive curvature disappears at hyperparameter extremes. It is removed by saturated softmax on one end (resulting in a lazy learning regime) and by vanishing logit gradients on the other (resulting in divergence). While the Goldilocks zone was originally linked to initialization scale by Fort & Scherlis (2019), Vysogorets et al. show it is equally achievable via softmax temperature tuning, which has an equivalent effect on loss curvature in homogeneous networks. In our HPO setting, varying learning rate and weight decay appear to traverse analogous optimization conditions, thereby inducing comparable transitions between regimes. From this perspective, Eq. (9) should be viewed as a phenomenological model summarizing how train and test losses behave across these regimes
>
> **Comparison with Vysogorets et al. optimization regimes.** To empirically support this connection, we replicated the exact setup used by Vysogorets et al. in their Section 5 experiment, using their original code. Precisely, we trained a homogeneous feedforward LeNet300-100 with full gradient descent on FashionMNIST for 20,000 iterations, sweeping learning rate and initialization scale. We compared this setup directly to one of our own dataset-model configurations, i.e., ResNet50 on CUB, sweeping learning rate and weight decay. As shown in Figure 2 of the revised manuscript:
>
> - Both setups exhibit very similar staircase-like heatmaps for training and testing accuracy, indicating sharp transitions in trainability consistent with the regime structure described by Vysogorets et al.
> - When plotting train vs test losses the replicated setup yields a qualitatively similar U-shaped relationship, closely matching the functional form of Eq. (9).
>
> These results suggest that the scaling law we observe is not an artifact of a specific architecture or dataset, but rather reflects how optimization dynamics organize homogeneous classifiers trained with cross-entropy loss across training regimes.
>
> **Broader empirical validation of the power law form.** Beyond this controlled comparison, we provide extensive empirical validation across our 37 dataset–model configurations. In the revised manuscript, Figure 10 (Appendix B) shows qualitative fits of the scaling law, and Figure 7 (Appendix B) reports quantitative goodness-of-fit, with a global $R^{2}$ of 0.896.
> This experimental evaluation spans a vast array of dataset domains (e.g., from natural to medical images) and data types (e.g., RGB and multi-channel images), a wide range of dataset (from tens to thousands of images per class) and model sizes (from few thousands to ~90M trainable parameters), and varying data augmentation strengths (from simple flipping to RandAugment). We emphasize that this does not establish universality, but demonstrates that the observed train–test relation is remarkably consistent across a wide range of practical deep learning settings.
>
> **Power law forms in recent literature.** Finally, our choice of a power-law parameterization is supported by recent work on loss-to-loss scaling (Brandfonbrener et al. 2025; Mayilvahanan et al. 2025). These empirical scaling-law studies confirm that test loss can be expressed as a power-law function of train loss.
>
> Fort, Stanislav, and Adam Scherlis. "The goldilocks zone: Towards better understanding of neural network loss landscapes.", AAAI 2019
>
> Vysogorets, Artem, et al. "Deconstructing the goldilocks zone of neural network initialization.", ICML 2024
>
> Brandfonbrener, David, et al. "Loss-to-loss prediction: Scaling laws for all datasets”, TMLR 2025
>
> Mayilvahanan, Prasanna, et al. "Llms on the line: Data determines loss-to-loss scaling laws”, ICML 2025

---

> ### Author Response · Authors · 2026-04-13
>
> > About their practical implementation
>
> We appreciate the reviewer’s attention to the separability threshold, as it is indeed a central component of our validation-free formulation. However, we believe there is a misunderstanding regarding its theoretical role and practical implementation. We therefore clarify two key points below, and subsequently provide additional empirical evidence demonstrating that the chosen threshold remains robust across unseen scenarios.
>
> **Our theoretical model does not strictly predict that the separability threshold must correspond to 100% training accuracy.** This point is mentioned in the last paragraph of our analysis of Section 3.2 “Practical approximations“, with full details in the original Appendix B (revised paper, Appendix C). We further discuss it in the sensitivity analysis, paragraph “Separability threshold”. The key reason is that our theory is formulated in terms of the differentiable smoothed normalized margin $\tilde{\gamma}$, rather than the true normalized margin $\bar{\gamma}$. These two quantities are related through the bound (Section 3.1):
>
> $\tilde{\gamma}(\theta) \ \leq \ \bar{\gamma}(\theta) \ \leq \ \tilde{\gamma}(\theta) + \frac{\log N}{\rho^k}$
>
> This implies that $\bar{\gamma}$ can already be positive while $\tilde{\gamma}$ is still negative, due to the finite gap $\frac{\log N}{\rho^k}$. In terms of classification accuracy, $\bar{\gamma} > 0$ corresponds to perfect separability (accuracy $=100%$), whereas $\tilde{\gamma} < 0$ indicates that the smoothed surrogate has not yet crossed the separability boundary. Conversely, $\tilde{\gamma} > 0$ is sufficient to guarantee $\bar{\gamma} > 0$, but the practical issue is that the converse need not hold.
>
> Therefore, the transition predicted by our theory occurs in a small neighborhood around the separability boundary, rather than exactly at 100% accuracy. Enforcing a strict 100% threshold is thus overly restrictive, whereas a slightly relaxed threshold (e.g., 99%) provides a practical proxy for detecting entry into the theoretically relevant separable regime.
>
> **Our sensitivity analysis in Section 4.3 empirically supports this prediction:** performance improves as the threshold approaches 100% (e.g., $\tau_{\text{acc}} = 99%$), but degrades when enforcing the strict 100% criterion. This behavior follows the theoretical argument above, as a hard 100% requirement can fail to capture configurations that already lie within the relevant near-separable regime, while still being indistinguishable from the boundary when viewed through the smoothed surrogate. We therefore emphasize that this analysis should not be interpreted as a meta-optimization of the threshold, but rather as an empirical validation of the theoretically predicted transition near the separability boundary.
>
> **Additional empirical evidence.** To further support the robustness of the separability threshold beyond our experimental setup, we additionally apply Twin to the setting of Vysogorets et al., which differs substantially from ours. In this case, the model is a feedforward LeNet300-100 trained with full gradient descent, without any scheduler or data augmentation, on FashionMNIST. Moreover, instead of sweeping weight decay, the study varies the initialization scale. Despite these differences, Twin accurately predicts the close-to-optimal configuration, achieving 89.18% test accuracy compared to the Oracle’s 89.41%. As shown in Figure 1 of the revised manuscript (top-left, top-central panels), the accuracy distribution of LeNet300-100 is long-tailed, further highlighting the non-triviality of the task and the robustness of the proposed threshold.
>
> Vysogorets, Artem, et al. "Deconstructing the goldilocks zone of neural network initialization.", ICML 2024

---

> > ### Author Response · Authors · 2026-04-13
> >
> > > Moreover, the applicability is limited to classification models; cannot cover LLM training scenarios.
> >
> > We agree that the current formulation is primarily tailored to classification settings. This choice is intentional, as our analysis builds on the established connection between margin, parameter norm, and cross-entropy loss in homogeneous classifiers from Lyu and Li (2019), which provides the theoretical basis for our theory enabling validation-free prediction.
> >
> > That said, we note that LLMs are also trained with cross-entropy objectives and are typically based on transformer architectures, which are compatible with the architectural settings considered in our analysis. From this perspective, Twin could in principle extend to LR and WD tuning. We further hypothesize that, since LLMs are commonly trained in single-pass or few-epoch regimes over very large datasets, they are unlikely to reach near-perfect training accuracy. As a result, optimization may predominantly remain in non-separable regimes, where the train–test scaling law is expected to be monotonic and the training loss remains predictive of test performance. However, we acknowledge that this extension is not explicitly validated in the current work, as it would have required large-scale HP sweeps that are computationally prohibitive for our resources.
> >
> > Finally, we emphasize that our primary motivation is settings where validation data is limited or expensive. In contrast, LLM training typically operates in regimes where abundant data allows for standard validation-based model selection, making our approach less directly applicable in that context. Exploring extensions to LLM-specific regimes remains an interesting direction for future work.

---

### Review · Reviewer_bJrY · 2026-03-30

**Summary Of Contributions:**

The authors propose a method for choosing hyper parameters without using a separate validation set, by proposing surrogate criteria that predict generalization (test loss).

From theory (and with some assumptions), they argue that the most promising hyper-parameters are those that led to the lowest training score if no run led to vanishing train error; but if some runs get training error below a small threshold, then those that led to the smallest weight norm should be preferred.

They provide various experiments that support the theory, then show that their method is competitive with other methods that do require a validation set.

**Additional Comments:**

I must stress that I am not an expert in the area. I could more or less follow the mathematical derivations, but I cannot vouch for their correctness. Similarly, while I appreciated the extensive review of related work, I cannot guarantee that it is exhaustive.

**Audience:**

Yes

**Audience Explanation:**

Being able to predict generalization without a test set is an intriguing objective. The possibility of such a prediction is certainly interesting in itself.

Practically, the "cost" of a separate validation set will usually be rather small, especially balanced against the need to trust a novel objective whose assumptions may or may not hold for the situation at hand. However, it may be useful in some situations. Besides, the theory developed is interesting in its own right.

**Broader Impact Concerns:**

I see no broader impact concerns.

**Claims And Evidence:**

Yes

**Claims Explanation:**

I found the experiments informative. The approximations involved in the theoretical derivations are clearly enunciated.

**Requested Changes:**

I have very few changes to propose. The paper is well-written.

In Figure 2, the top row has a green dot and green line. What does it mean exactly? Can we have the corresponding data point in the bottom row also colored in green, which would facilitate comparison?

---

> ### Author Response · Authors · 2026-04-13
>
> We thank the reviewer for the time invested in reviewing our paper and for the thoughtful and positive evaluation of our work. We are glad that you found the paper well-written, the experiments informative, and the overall idea interesting and potentially useful to a broad audience.
>
> We also appreciate your careful reading and helpful suggestion regarding Figure 2. In response, we have removed the green line to avoid potential confusions since expressing its meaning, i.e., the found parameter C, in a clear manner was an actually difficult task considering the limited space available in the figure.

---

### Author Response · Authors · 2026-04-13

We thank all the reviewers for the time they invested in evaluating our paper and for their constructive feedback. Below, we summarize the strengths, our revisions, and clarifications. We have updated the revised manuscript (cancelled red text means removed text and underlined blue means additions).

**Strengths**

S1. Reviewers bJrY and ZyEs highlighted the importance of predicting generalization without validation data, especially in data-constrained scenarios.

S2. Reviewers noted that Twin consistently matches "Oracle" performance (test-set tuning) across 37 configurations.

S3. The manuscript was praised for its clear presentation and informative visualizations.

**Revisions**

R1. Scaling Law Justification (Reviewer Bn2y, ZyEs): We expanded Section 3 to emphasize that Eq. 9 is an empirical scaling law. We added a discussion connecting this to the Goldilocks zone (Vysogorets et al., 2024) and included new results in Figure 2 showing our U-shaped test-loss relationship aligns with these optimization regimes. We further added qualitative and quantitative empirical validations of Eq. 9 in Appendix B.

R2. Related Work on Scaling Laws: To address concerns regarding the power-law form, we added references to Brandfonbrener et al. (2025) and Mayilvahanan et al. (2025), which establish that test loss consistently follows shifted power laws relative to training loss across diverse architectures and datasets.

R3. Technical Corrections: We updated Algorithm 1 to use Boolean indexing instead of a Hadamard product for filtering configurations. We also added Figure 11 to the Appendix to visualize the high sensitivity and difficulty of the tuning tasks across evaluated models.

R4. Structural Flow: Added a roadmap at the start of Section 4 to clarify the three distinct experimental phases.

**Clarifications**

C1. Scope: While Twin's foundation is tied to exponential-tailed losses (cross-entropy), we clarify in Appendix F that extending this to L2 regression or other HP sweeps in different contexts is a target for future work.

C2. Terminology: We defined "trivial accuracy" as random-chance performance and clarified that the 15% training floor is a heuristic to exclude collapsed runs.

C3. HPO Settings: We clarified that "FIFO" is standard nomenclature for sequential scheduling without early stopping. The varied trial counts (36 vs 100) demonstrate Twin's robustness to different search space budgets.

C4. Early Stopping: Clarified that we utilize standard HyperBand logic applied to training loss, rather than custom-tuned validation metrics.

C5. Separability Threshold (Reviewer Bn2y): We referenced Section 3.2 (+ Appendix) and Section 4.3 to explain that the 99% threshold is a practical proxy for the smoothed margin regime, which occurs before perfect 100% separability due to the finite gap between smoothed and true normalized margins.

---

### Decision · Action_Editor_BF5z · 2026-05-06

**Recommendation:** Accept with minor revision

**Additional Comments:**

Two reviewers indicated Leaning Accept and one Leaning Reject. After reading the comments and discussions, my recommendation is Accept with minor revision, provided that the authors address the remaining concern as follows.

**Revision request:**
- The current version (after the revision during the discussion period) introduces Eq. (9) in a way that may not be immediately clear on the point that the relationship is an assumption based on empirical observations. To avoid any potential misunderstanding, the paper should more clearly and concisely state, perhaps right before or after Eq. (9) is introduced, that the relationship is an assumption that the authors hypothesize based on empirical observations and there is no conclusive evidence yet that it holds accurately in all cases. (Indeed, the U curves do not fit well in a few cases of Figure 10.)

- Some model selection methods such as those based on AIC, BIC, and MDL might be related and I encourage the authors to add a discussion if relevant.

- I advise the authors to follow the Math section of this guide:
https://www.jmlr.org/format/formatting-errors.html

**Audience:**

Yes

**Audience Explanation:**

Hyper-parameter selection without a validation set is a relevant topic of research. The reviewers appreciate the empirical results and agree that the paper is likely to interest some of TMLR's audience.

**Claims And Evidence:**

Yes

**Claims Explanation:**

The reviewers mentioned that the effectiveness of the proposed method are well demonstrated by the empirical results.

Two of the reviewers raised concerns about the insufficient justification of the model assumed on the relationship between the training and the test risks (Eq. 9). The authors added empirical results (Figure 2 and Appendix B) to verify the relationship and added paragraphs including a connection to the work by Vysogorets et al. (2024).
While they address the issue to some extent, a reviewer expressed that Eq. (9) is still not completely supported, and it might still mislead the readers.

However, I believe that the authors can make this point clearer with a minor revision. I describe this minor revision request more in detail in the Additional Comments.